# FULLY IDENTICAL INITIALIZATION

## ABSTRACT

Deep neural networks (DNNs) have achieved numerous remarkable accomplishments in practice. The success of these networks hinges on effective initialization methods, which are vital for ensuring stable and rapid convergence during training. Recently, initialization methods that maintain identity transition within layers have shown good efficiency in network training. These techniques (e.g., Fixup) set specific weights to zero to achieve identity control. However, settings of remaining weight (e.g., Fixup uses random values to initialize non-zero weights) will affect inductive bias that is achieved only by a zero weight, which may be harmful to training. Addressing this concern, we introduce fully identical initialization (IDInit), an innovative method that preserves identity in both the main and substem layers of residual networks. IDInit employs a padded identity-like matrix to overcome rank constraints in non-square weight matrices. Furthermore, we show a convergence problem of an identity matrix can be solved by adding a momentum term into the optimizer. Additionally, we explore enhancing the universality of IDInit by processing higher-order weights and addressing dead neuron problems. IDInit is a straightforward yet effective initialization method, promising improved convergence, stability, and performance across various settings, including large-scale datasets and deep models. It stands as a novel solution for initializing non-standard weight matrices, offering significant advantages in network training.

## 1 INTRODUCTION

Deep neural networks (DNNs) have attracted significant attention due to their versatility in various applications (He et al., 2016; Li et al., 2021). Behind these successes, initialization methods play a crucial role in promoting stable and fast-convergent training processes for networks (Sutskever et al., 2013; Arpit et al., 2019; Huang et al., 2020; Pan et al., 2022). Usually, initialization methods make effects by controlling the magnitude of signals. For example, Xavier (Glorot & Bengio, 2010) initialization is originally proposed to maintain signals in the non-saturated region of the sigmoid activation function by restricting signal variances, which greatly solved the difficulty of training. Then, Gilboa et al. (2019); Poole et al. (2016) propose to initialize network weights by constraining signals on the edge of chaos through dynamical isometry, which can further benefit the network training. Later, Hardt & Ma (2017) analyzed the optimization landscape of linear residual networks, and found that weights that transit identity in layers can help networks converge fast as their F-norm is close to that of the final converged weights. And identity transition also corresponds to isometry theory (Zhang et al., 2019), thereby, contributing to avoiding gradient explosion and diffusion.

An instance of preserving identity across neural network layers, known as "identity-control," is depicted in Figure 1 and formally expressed as $Y = X$. This type of initialization can be implemented by setting specific weights (e.g., $W_2$) to $\mathbf{0}$, thereby ensuring zero output in the sub-stem, as elucidated by Hardt & Ma (2017). This approach, however, poses challenges in configuring the remaining weight $W_1$. Previous work such as Fixup (Zhang et al., 2019) and ZerO (Zhao et al., 2021) initialize $W_1$ using the Xavier and Hadamard methods, respectively. These initializations can adversely affect the inductive bias already established by setting $W_2 = \mathbf{0}$, a setting beneficial for training. As evidenced in Figure 2, both Xavier and Hadamard methods cause

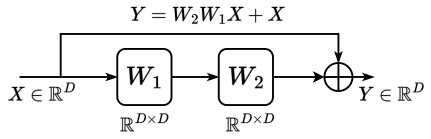

$$Y = W_2 W_1 X + X$$

Identity-Control: $Y = X$

Figure 1: A case of identity-control initialization. In default, $W_1$ and $W_2$ are usually initialized with the Xavier method. Identity transition is achieved if $W_2 = \mathbf{0}$.

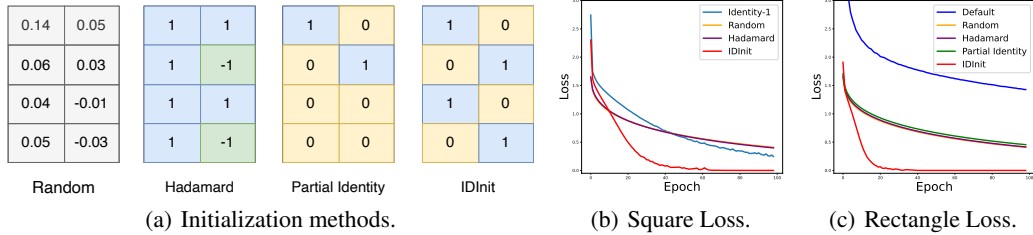

(a) Initialization methods.      (b) Square Loss.      (c) Rectangle Loss.

Figure 2: Analyzing effect of initializing $W_1$ while $W_2 = \mathbf{0}$. The experiment uses Cifar10 and blocks in Figure 1, and more details are in Appendix C.4. (a) The initialization methods for $W_1$ in a rectangular format. Fixup: "Random"; ZerO: "Hadamard". And "Partial Identity" and "IDInit" denote padding $\mathbf{0}$ and $I$ to an identity matrix, respectively. (b) Set $W_1 \in \mathbb{R}^{240 \times 240}$ and $W_2 \in \mathbb{R}^{240 \times 240}$ as square matrices. "Identity-1" represents a configuration where only one weight is initialized as $\mathbf{0}$. Interestingly, while "Random" and "Hadamard" methods may outperform "Identity-1" in initial training epochs due to more network weights, they are hard to capture the inductive bias of "Identity-1", resulting in convergence difficulties. In contrast, IDInit can effectively leverage the training dynamics associated with "Identity-1". (c) Set $W_1 \in \mathbb{R}^{280 \times 240}$ and $W_2 \in \mathbb{R}^{240 \times 280}$ as rectangle matrices. "Default" means $W_1$ and $W_2$ are initialized with Xavier. However, "Default" proves ineffective for training, as it conflicts with dynamical isometry. Furthermore, even though "Partial Identity" exhibits the capability to transmit partial signals, it performs poorly due to rank constraint issues. Finally, IDInit maintains well-training conditions by padding the identity matrix.

difficulties in achieving convergence. Observing this, we propose initializing $W_1$ with an identity matrix $I$, which retains the inductive bias as $IW_2 \equiv W_2$. Moreover, $I$ also achieves dynamical isometry in the sub-stem layer as discussed by Zhao et al. (2021). Figure 2 demonstrates that using an identity matrix significantly aids in training convergence. Nonetheless, the practical application of an identity matrix faces two primary obstacles. First, an identity matrix requires square-shaped weights, a condition seldom met in practical networks. While a partial identity matrix (by padding $\mathbf{0}$ to an identity matrix) offers a workaround, it leads to rank constraints issues (Zhao et al., 2021) when the output dimension exceeds the input dimension, impairing network generalization. The second obstacle concerns the convergence capability. As Bartlett et al. (2019) pointed out, weights initialized with an identity matrix are difficult to converge to the ground truth, of which eigenvalues contain negative values. This convergence problem is important as it indicates a limited universality of applying an identity matrix as an initialization method.

**IDInit.** In light of the preceding discussion, we are going to address the two major obstacles. To handle a non-square matrix, we pad a new identity matrix in adjacency to an identity matrix. We theoretically demonstrate this operation can resolve the rank constraint problem. Then, to alleviate the replica problem induced by this padding scheme, we impose a loosening condition on the padded identity-like matrix. Turning to the matter of convergence, we conduct an experiment to analyze it. Interestingly, we find that the convergence problem can be solved by adding a moment in an optimizer (e.g., the stochastic gradient descent optimizer), which is the most general setting for training neural networks. By introducing the identity-like matrix into the identity-control framework, we implement a fully identical initialization (IDInit), which ensures identity transition across both main and sub-stem layers. Moreover, we explore two additional techniques for improving the universality of IDInit and the identity-control framework:

(1) Higher-order Weights: An identity matrix is a 2-D array and it is necessary to consider an efficient method to transfer the identity matrix to a higher-order weight (e.g., a 4-D convolution). A previous strategy is to keep identity along the channel (see Sec. 3.3.1). However, this method causes diversity loss in channels, which is harmful to performance. To remedy this shortage, we propose to keep identity in patches alternatively for more diversity in channels to achieve improvement.

(2) Dead Neurons: As an identity-control method, IDInit sets the last layer of the sub-stem to 0 for transiting identity in the main branch. However, a dead neuron problem is possibly caused by directly this setting, especially for residual convolutional neural networks, (Zhang et al., 2019;

Zhao et al., 2021). Addressing this, we select some elements to a small numerical value $\varepsilon$ to increase trainable neurons as in Figure 6.

To our knowledge, IDInit is the first successful trial to maintain identity in both main- and sub-stems by breaking the rank constraints, which promise the expressive power of IDInit. Then, we address the replica problem by adding small noise while maintaining the dynamical isometry. By further proposing modifications to CNNs and solutions to dead neuron problems, we have significantly improved accuracies by 3.42% and 5.89% respectively (see Section 4.4). Note that, although the identity matrix is used as initialization in prior work, it was only used for square matrix, e.g., Le et al. (2015) set a hidden-to-hidden layer in a recurrent neural network with an identity matrix for better performance. IDInit is novel for the consideration of non-standard situations, e.g., non-square matrix. On ImageNet, IDInit can achieve almost all the best performance (with an average 0.55% improvement) and the fastest convergence on various settings. IDInit can accelerate the training procedure of BERT-Base, manifesting an 11.3% reduction in computational cost. Therefore, our approach yields consistently significant advantages in the training of neural networks.

## 2 RELATED WORK

Given an $L$-layer residual network with two parameters in each residual stem and an input signal $x^{(0)}$, the $i$-th layer can be formulated as

$$x^{(i+1)} = a(I + \theta^{(i,0)}\theta^{(i,1)})x^{(i)}, \tag{1}$$

where $a(\cdot)$ denotes the activation function, $x^{(i)}$ means an input of $i$-th residual block in a network, $I$ is an identity matrix denoting residual connection, and $\theta^{(i,0)}$ and $\theta^{(i,1)}$ are weights in the $i$-th residual stem of a residual block.

**Dynamical Isometry.** Assuming signal magnitude (e.g., $\sigma^2(x^{(i)})$) of each layer changing in a scale $\alpha$, the last signal magnitude can reach $\alpha^L$ (e.g., $\sigma^2(x^{(L)}) = \alpha^L\sigma^2(x^{(0)})$), making it easy to cause signal explosion and diffusion, especially for large $L$. Dynamic isometry is also a mechanism that comes from mean-field theory (Pennington et al., 2017; 2018). By utilizing this paradigm, it is usually to consider the input-output Jacobian

$$J_{io} = \frac{\partial x^{(L)}}{\partial x^{(0)}}, \tag{2}$$

whose mean squared singular value is $\chi$. (Pennington et al., 2017) and (Bachlechner et al., 2021) show that $\chi > 1$ indicates that the model is in a chaotic phase, and back-propagated gradients will explode exponentially. By contrast, $\chi < 1$ means a model in an ordered manner that back-propagated gradients vanish exponentially. $\chi = 1$ is a critical line of initialization, avoiding vanishing or exploding gradients. The isometry can provide sufficient robustness for the network training (Gilboa et al., 2019; Poole et al., 2016; Yang & Schoenholz, 2017).

**Network Initialization.** Common initialization methods are Xavier (Glorot & Bengio, 2010) and Kaiming initialization (He et al., 2015). Especially for residual networks efficiency, Hardt & Ma (2017) theoretically demonstrates that network training benefits from keeping identity. Le et al. (2015) set a hidden-to-hidden layer in a recurrent neural network with an identity matrix for better performance. Fixup (Zhang et al., 2019) and ZerO (Zhao et al., 2021) both set residual stem to 0 (not residual connections) to guarantee the identity of signals, thereby initializing ResNets successfully. SkipInit (De & Smith, 2020) replaces Batch Normalization with a multiplier whose value is 0. ReZero (Bachlechner et al., 2021) directly adds extra parameters of value 0 to keep identity, leading to fast convergence.

**Identity-Control Training Framework.** Net2Net (Chen et al., 2016) proposes to expand network depth by maintaining identity. DiracNet (Zagoruyko & Komodakis, 2017) maintains an identity for propagating information deeper into the network. However, it suffers from reducing residual connection, causing performance loss. ISONet (Qi et al., 2020) is an isometric learning framework that contains an identical initialization (i.e., the Dirac function that is also used in ZerO (Zhao et al., 2021) by padding 0 in a non-square matrix case), and isometric regulation in training. ISONet multiplies 0 to the residual stem like Fixup (Zhang et al., 2019). ISONet lacks the flexibility for various convolutions as it specifies the net without normalization, and requires SReLU.

## 3 IDENTICAL INITIALIZATION

In this section, we first discuss the convergence problem in Sec. 3.1. Then, we describe the way to maintain the stability for non-square identity-like matrices in Sec. 3.2. At last, we introduce an efficient reshaping strategy to significantly enhance convolution performance and the method to tackle the dead neuron problem in identity transition in Sec. 3.3.1 and Sec. 3.3.2, respectively.

### 3.1 CONVERGENCE ABILITY OF IDINIT

To address the convergence problem proposed by Bartlett et al. (2019), we conduct an experiment to elaborate on this problem. We set the target matrix as $-I \in \mathbb{R}^{10 \times 10}$. We use a 3-layer net, weights in which are $W_0, W_1, W_2 \in \mathbb{R}^{10 \times 10}$. We randomly generated 4000 data pairs $\{X_i, Y_i\}$ by $Y_i = -I X_i + \xi$, where $X_i, Y_i \in \mathbb{R}^{10}$, and $\xi$ is noise with mean 0 and std 1e-2. We use 2000 samples for solving the least square (LS) matrix or training the network. We use the other 2000 samples for testing. Mean squared error (MSE) is used as the loss function.

Table 1: Results of the convergence problem.

| Method | Solution w/o Momentum | w/ Momentum |
|---|---|---|---|
| LS | 1e-4 | - | - |
| Xavier | - | 1e-4 | 1e-4 |
| Identity | - | 1 | 1e-4 |

As shown in Table 1, SGD w/o momentum is hard to optimize a network initialized with the identity matrix, which corresponds to Bartlett et al. (2019). However, when used in conjunction with momentum, SGD is able to successfully optimize the network. It's worth noting that while a network initialized with Xavier can be optimized by SGD without momentum, wider and deeper networks still cannot be well optimized under the same conditions (Sutskever et al., 2013). As such, momentum is crucial in training deep networks. In conclusion, although IDInit may have trouble achieving convergence with pure SGD, it can be optimized effectively with momentum, which is the most commonly used setting.

### 3.2 MAINTAINING IDENTITY BY PADDING IDENTITY

A standard identity matrix can naturally satisfy identity transition. However, in a non-square situation, this natural advantage is lost. To address this problem, we pad the identity matrix on an identity matrix to fit a non-square matrix. Specifically, for a fully-connected layer transformed from Eq. (1) as $x^{(i+1)} = \theta^{(i)} x^{(i)}$, we set the weight $\theta^{(i)} \in \mathbb{R}^{D_{i+1} \times D_i}$ to

$$\theta^{(i)}_{m,j} = \begin{cases} \tau, & \text{if } m \equiv j \pmod{D_i}, \\ 0, & \text{otherwise.} \end{cases} \quad (3)$$

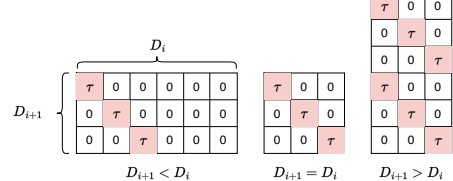

Figure 3: An overview of $IDI_\tau$. $D_i$ means an input dimension and $D_{i+1}$ denotes an output dimension. $\tau$ is usually set to 1 to maintain identity.

The initialization formulated as Eq. (3) is termed as $IDI_\tau$, where IDI means the identical initialization function, and $\tau$ is calculated by considering the activation function, e.g., $\tau_{ReLU} = \frac{1}{2}$ and $\tau_{sigmoid} = 1$ for ReLU and sigmoid activation functions respectively. As in Figure 3, setting $\tau = 1$ can form $IDI_1$ initialization. Moreover, we provide a trainability analysis on the non-squared condition of $IDI_\tau$ in Sec. A.2 of the appendix.

#### 3.2.1 ON RANK CONSTRAINT PROBLEM

ZerO (Zhao et al., 2021) points out that a dimension-increasing matrix (i.e., $D_{i+1} > D_i$) can encounter a rank constraint problem if padding zero values.

**Rank Constraint Problem.** Consider an $L$-layer network with the formulation of the $i$-th layer ($i \in \{0, 1, \ldots, L-1\}$) as $x^{(i+1)} = \theta^{(i)} x^{(i)}$, where $x^{(0)} \in \mathbb{R}^{D_0}$, $x^{(0)} \in \mathbb{R}^{D_L}$, $\theta^{(0)} \in \mathbb{R}^{D_h \times D_0}$, $\theta^{(L-1)} \in \mathbb{R}^{D_L \times D_h}$, $\theta^{(k)} \in \mathbb{R}^{D_h \times D_h}$ where $k \in \{1, 2, \ldots, L-2\}$, and $D_h > D_0, D_L$. Define residual component $\hat{\theta}^{(k)} = \theta^{(k)} - I$. When initializing the dimension-increasing weight $\theta^{(0)}$ by padding zeros values, the rank constraint problem is performed as

$$\text{rank}(\hat{\theta}^{(k)}) \leq D_0, \quad (4)$$

where $k \in \{1, 2, \ldots, L-2\}$. $\mathrm{IDI}_\tau$ can simply break the rank constraint as in Theorem 3.1. We defer the proof in Appendix A.3.

**Theorem 3.1.** *If initializing all weights $\{\theta^{(i)}\}_{i=0}^{L-1}$ with $\mathrm{IDI}_1$, the rank of middle layers can attain*

$$rank(\hat{\theta}^{(k)}) \geq D_0, \quad where \quad k \in \{1, 2, \ldots, L-2\}, \tag{5}$$

*which breaks the rank constraint.*

Notably, ZerO (Zhao et al., 2021) claimed that ranks of weights are always limited to $D_0$ when the non-linearity like ReLU is not applied, which seems to contradict Theorem 3.1. However, this claim is tenable in the initial state. After training for several steps, an IDInit initialized network can break this constraint.

**Replica Problem.** When recurrently padding the identity matrix, the output features are still replicated. According to Blumenfeld et al. (2020), such a replica problem can be solved by adding noise to weights. Inspired by that, we loosen the identity condition to generate $\tau \sim N(\tau, \epsilon_\tau)$, while keeping most identity. $\epsilon_\tau$ is a small value and set to 1e-6 in this paper. With this loose condition, IDInit can give additional noise to output features and bring more feature diversity. Profiting from the feature diversity, IDInit therefore can increase the rank values as shown in Figure 4(b).

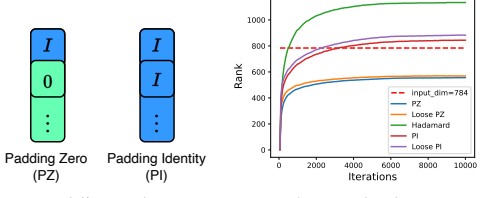

(a) Padding schemes.    (b) Rank plot.

Figure 4: Two padding schemes and their influence on ranks of a layer. We trained a 3-layer network on MNIST, and set $D_0 = 768$ and $D_h = 2048$. We plot $rank(\theta^{(1)}) \in \mathbb{R}^{D_h \times D_h}$ in (b). As shown in (b), padding identity can achieve more than a rank of 768 like Hadamard, while padding zero is limited under 768. The loose condition can lead to better rank performance, however, cannot solve the rank constraint problem of padding zero.

### 3.3 ADDITIONAL TECHNIQUES FOR IDINIT

In this part, we introduce two techniques to enhance the universality of IDInit for practical situations.

#### 3.3.1 PATCH-MAINTAIN CONVOLUTION

Convolution layers are important structures in deep neural networks. Here, we will explore an initialization pattern for convolution with identity transition. A convolution kernel is usually defined as $\mathcal{C} \in \mathbb{R}^{k \times k \times c_{in} \times c_{out}}$, where $c_{in}$ and $c_{out}$ are channels of input and output separately, and $k$ means convolutional kernel size. Similar to an identity matrix, Zhao et al. (2021) consider a convolution layer that transits identity by setting 0-filled $\mathcal{C}$ through $\mathrm{IDI}_\tau(\mathcal{C}_{n,n,:,:})$, where $n \in \mathbb{N}^+$ and $k = 2n+1$. As a convolutional kernel window size, $k$ is usually an odd number. When $c_{in} = c_{out}$, the convolution maintains the identity. When $c_{in} > c_{out}$ or $c_{in} < c_{out}$, $\mathcal{C}$ will under-sample and over-sample on an input feature along channel respectively. Keeping identity is usually considered an efficient way to improve model performance, however, we find that this setting can lead to a fatal performance degeneration (see Sec. 4.4).

**Patch-Maintain Convolution.** Inspired by Han et al. (2020) that enhance model performance by increasing feature diversity, we propose to fuse spatial information by simply reshaping a matrix initialized with $\mathrm{IDI}_\tau$. Specifically, we reshape the convolutional kernel $\mathcal{C}$ into a matrix $C \in \mathbb{R}^{c_{out} \times kkc_{in}}$. We initialize $C$ as

$$\mathrm{IDI}_\tau(C). \tag{6}$$

Then by reshaping $C$ into $\mathcal{C} \in \mathbb{R}^{k \times k \times c_{in} \times c_{out}}$, our initialization for a convolution is completed. This reshaping strategy can shift spatial features, thereby increasing feature diversity. We utilize $\mathrm{IDIC}_\tau$ to denote such a reshaping process. A detailed description is in Figure 11 in the Appendix.

#### 3.3.2 TACKLING DEAD NEURONS

At present, residual blocks become the most popular module in almost all the neural network (e.g., Mixers (Liu et al., 2021a; Tolstikhin et al., 2021), Convolutions (He et al., 2016; Zhang et al., 2019),

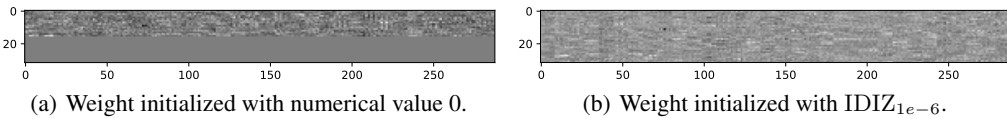

(a) Weight initialized with numerical value 0.  (b) Weight initialized with $\mathrm{IDIZ}_{1e-6}$.

Figure 5: The last weight in a residual block of a trained ResNet. More than half of elements in (a) are not trained, which is known as the dead neuron. By contrast, $\mathrm{IDIZ}_{1e-6}$ successfully solves the dead neuron problem and makes all the elements in (b) trainable.

and Transformers (Liu et al., 2021b; Vaswani et al., 2017)). A residual block is usually constructed with a residual connection and several transformations in the residual stem.

Given a residual network formulated by Eq. (1), recent research (Zhang et al., 2019; Zhao et al., 2021) directly sets the last transformation in the residual stem to 0, i.e., $\theta^{(i,0)} = 0$, thereby maintaining an identity as

$$x^{(i+1)} = (I + 0)x^{(i)} = x^{(i)}. \tag{7}$$

However, the setting can possibly cause a dead neuron problem.

**Dead Neuron Problem.** Fixup (Zhang et al., 2019) only uses a multiplier of 1 after $\theta^{(i,0)} = 0$, thereby obtaining non-zero gradients. However, in a realistic implementation of neural networks, the multiplier of Batch Normalization can be set to 0 (Goyal et al., 2017), and down-sampling operation can also cause 0 filled features[1][2]. Under the implementations, $\theta^{(i,0)}$ always acquires gradients with 0 values, known as the dead neuron problem, which causes failed weight updating.

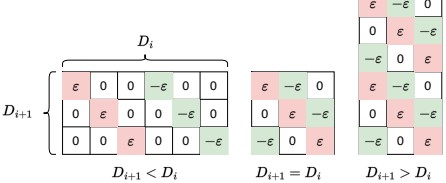

Figure 6: An overview of $\mathrm{IDIZ}_\varepsilon$. $D_i$ means an input dimension and $D_{i+1}$ denotes an output dimension. We set $\epsilon$ to 1e-6 to transit zero.

Tackling this problem, we generate small values on $\theta^{(i,0)}$ to assist in training. Recall the goal of identity-control initialization that outputs 0. Therefore, we build a calculation to get the expectation and variance of outputs approaching 0. Considering two i.i.d variables, $v_1$ and $v_2$, whose variances are $\sigma^2(v_1) = \sigma^2(v_2) = \varphi$ and means are $\mu(v_1) = \mu(v_2) = \gamma$, the variable $v = \varepsilon(v1 - v2)$ have

$$\begin{cases} \mu(v) = 0, \\ \sigma^2(v) = 2\varphi\varepsilon^2, \end{cases} \tag{8}$$

where $\varepsilon$ is a coefficient, and $\sigma^2(v)$ will be limited to 0 when $\varepsilon$ is sufficiently small. Assuming elements of $x^{(i)}$ are i.i.d to each other, by applying subtraction on any two elements, the result has a mean of 0, and a variance related to $\varepsilon$. We also take $\theta^{(i,0)} \in \mathbb{R}^{D_{i+1} \times D_i}$ as an instance. At first, we initialize $\theta^{(i,0)}$ with $\mathrm{IDI}_\varepsilon$. Then consider two cases: (i) if $D_{i+1} < D_i$, setting $\theta^{(i)}_{:,D_{i+1}+1:D_i}$ with $\mathrm{IDI}_{-\varepsilon}$; (ii) if $D_{i+1} \geq D_i$, set $\theta^{(i)}_{m,j} = -\varepsilon$, when $m\%D_i = j - 1$. Therefore, we can obtain a variance of 0 by setting $\varepsilon$ to a small value. This method is termed as $\mathrm{IDIZ}_\varepsilon$, and we illustrate some cases in Figure 6. In this paper, we set $\varepsilon = 1e - 6$ everywhere. As shown in Figure 5, $\mathrm{IDIZ}_{1e-6}$ successfully initializes the last weight in a residual block. In addition, we also transform $\mathrm{IDIZ}_\varepsilon$ to a convolution form $\mathrm{IDIZC}_\varepsilon$ through the patch-maintain scheme.

We conclude the whole IDInit as (1) **Non-Residual Networks.** Directly applying $\mathrm{IDI}_\tau$ and $\mathrm{IDIC}_\tau$ to all the fully-connected and convolutional layers, respectively; (2) **Residual Networks.** (i) Applying $\mathrm{IDI}_\tau$ and $\mathrm{IDIC}_\tau$ to all the fully-connected and convolutional layers, respectively; (ii) Applying $\mathrm{IDIZ}_\varepsilon$ and $\mathrm{IDIZC}_\varepsilon$ to the fully-connected and convolutional layers in the last position of residual blocks, and the position of last classification layer.

---

[1] https://github.com/hongyi-zhang/Fixup/blob/master/cifar/models/resnet_cifar.py
[2] https://github.com/akamaster/pytorch_resnet_cifar10/edit/master/resnet.py

## 4 EXPERIMENTS

In this section, we construct a set of experiments to validate the proposed IDInit. Firstly, we conduct experiments on non-residual convolution and residual convolution in Sec. 4.1 and Sec. 4.2, respectively. Then we conduct image classification on ImageNet in Sec. 4.3. Next, we implement an ablation experiment in Sec. 4.4 to show the effect of the proposed two modifications in Sec. 3. Later we conduct a text classification experiment in Sec. 4.5. At last, we employ a pre-training experiment on the large-scale dataset WuDaoCorpora in Sec. 4.6 separately. We also analyze the variance amplification in Sec. C.2, weight distribution in Sec. C.3, and dynamical isometry in Sec. C.4.

### 4.1 VALIDATION ON NON-RESIDUAL CONVOLUTION

We use this experiment to show IDInit can achieve a good initial state for training. In this experiment, we use Cifar10 as the benchmark dataset. We use nine convolutional layers named All-Conv (Springenberg et al., 2015). We show the structure of AllConv in Table 6 in the appendix. The optimizer is Stochastic Gradient Descent (SGD) with momentum 0.9, weight decay 5e-4, and learning rate 1e-1. The learning rate scheduler adopts a warm-up cosine reduction strategy. We run the model in 300 epochs on one Nvidia A100. We adopt Kaiming initialization and IDInit w/o $IDIC_\tau$ initialization for comparison.

Results are shown in Figure 7, without a warm-up strategy which is a strong trick for training, both Kaiming and IDInit w/o $IDIC_\varepsilon$ fail to train the model. By contrast, our initialization can train AllConv and maintain the highest performance in all situations, showing a strong effect on stability and performance. As IDInit w/o $IDIC_\varepsilon$ performs poorly, we demonstrate the patch-maintain

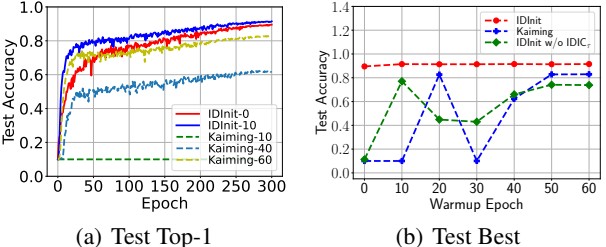

(a) Test Top-1      (b) Test Best

Figure 7: Results of AllConv on Cifar10. The number behind the initialization denotes the warm-up epochs.

strategy mentioned in Sec. 3.3.1 can be good for increasing feature diversity. This experiment shows the identical method can be a feasible initialization for non-residual networks.

Table 2: Results on Cifar10. ZerO performs worse for zero down-sampling as mentioned in Sec. 3.3.2. IDInit can always achieve fast convergence.

| | 56 Layer (SGD/Adam) | | 110 Layer (SGD/Adam) | |
|---|---|---|---|---|
| Initialization | Acc. | Epochs to 80% Acc. | Acc. | Epochs to 80% Acc. |
| Zero $\gamma$ | $92.32_{\pm0.19}/87.37_{\pm0.43}$ | $57_{\pm7}/63_{\pm4}$ | $93.07_{\pm0.28}/88.30_{\pm0.31}$ | $36_{\pm2}/56_{\pm7}$ |
| ZerO | $90.57_{\pm0.31}/83.53_{\pm0.42}$ | $57_{\pm3}/85_{\pm4}$ | $91.71_{\pm0.21}/84.24_{\pm0.10}$ | $55_{\pm3}/76_{\pm2}$ |
| Fixup | $93.24_{\pm0.82}/89.50_{\pm0.18}$ | $\underline{31}_{\pm3}/55_{\pm3}$ | $93.32_{\pm0.23}/90.67_{\pm0.12}$ | $33_{\pm3}/49_{\pm2}$ |
| SkipInit | $92.29_{\pm0.30}/85.45_{\pm0.74}$ | $\mathbf{26}_{\pm1}/81_{\pm3}$ | $92.67_{\pm0.16}/87.18_{\pm0.94}$ | $\underline{31}_{\pm5}/70_{\pm7}$ |
| ReZero | $93.06_{\pm0.54}/89.26_{\pm0.30}$ | $33_{\pm2}/\underline{44}_{\pm3}$ | $94.03_{\pm0.26}/90.25_{\pm0.20}$ | $35_{\pm5}/\underline{38}_{\pm3}$ |
| Kaiming | $93.36_{\pm0.14}/87.55_{\pm0.32}$ | $34_{\pm3}/50_{\pm2}$ | $94.06_{\pm0.18}/87.89_{\pm0.41}$ | $33_{\pm4}/56_{\pm3}$ |
| IDInit | $93.41_{\pm0.10}/90.01_{\pm0.32}$ | $\mathbf{26}_{\pm1}/\mathbf{34}_{\pm1}$ | $94.04_{\pm0.24}/90.53_{\pm0.10}$ | $\mathbf{27}_{\pm1}/\mathbf{36}_{\pm2}$ |

### 4.2 VALIDATION ON RESIDUAL CONVOLUTION

In this experiment, we validate the proposed initialization with the comparison with existing initialization, including (1) Fixup; (2) SkipInit; (3) ReZero; (4) Kaiming; (5) Zero $\gamma$ (Setting the scale in Batch Normalization (BN) to 0) (Goyal et al., 2017); (6) ZerO. We use ResNet-56/110 as backbones on Cifar10. To verify the universality, we use two settings, namely w/ and w/o BN. For analyzing convergence, we adopt both SGD and Adam optimizer for updating models. We set SGD, with the

momentum 0.9, the weight decay 5e-4, and the learning rate 0.2. For Adam, the learning rate is 0.001, $\beta_1$ is 0.9 and $\beta_2$ is 0.999. We train models for 200 epochs.

Results are shown in Table 2. Although ZerO uses the Hadamard matrix to break the rank constraint problem, it can be damaged by zero down-sampling as mentioned in Sec. 3.3.2. Therefore, we reclaim the importance of using $IDIZ_\varepsilon$ and $IDIZC_\varepsilon$ for avoiding such potential damage. Compared with baselines, IDInit derives the best accuracies in most cases. In addition, IDInit can achieve the least epochs to reach 80% accuracy in all settings, which shows a good convergence ability.

### 4.3 IMAGE CLASSIFICATION ON IMAGENET

We validate ViT-B/32 (Dosovitskiy et al., 2021), ResNet-50/152 (RN-50/152) and Se-ResNet-50 (SRN-50) as backbones on ImageNet in this experiment. For ViT-B/32, the optimizer is AdamW with a learning rate 1e-3 and a weight decay 5e-2. The training epochs is 300. We use 30 epochs for warm-up. For RN-50/152 and SRN-50, we use SGD with a learning rate 1e-1 and a weight decay 1e-4 for 90-epoch training. We use 9 epochs for warm-up. For all models, the batch size is 1024, and

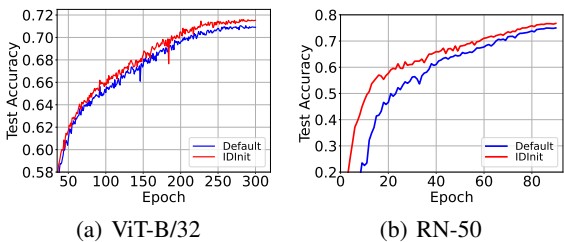

(a) ViT-B/32      (b) RN-50

Figure 8: Some results on ImageNet. "Default" means the default initialization of models.

we apply data augment including cutmix (Yun et al., 2019) with $\alpha = 1.0$, mixup (Zhang et al., 2018) with $\alpha = 0.8$, the switching probability is 0.5 and a label smoothing with 0.1.

Results are shown in Figure 8 and Table 3. On three types of networks, i.e., ViT, ResNet and Se-ResNet, and multiple depths, IDInit always achieves faster

Table 3: Results on ImageNet. The value in brackets means "Epochs to 60% Acc". On average, IDInit enhances accuracy by 0.55% compared to the baseline and expedites model convergence by 7.4 epochs.

| Model | ViT-B/32 | RN-50 (Adamw) | RN-50 | SRN-50 | RN-152 | Avg ($\Delta$) |
|---|---|---|---|---|---|---|
| Default | 71.05 (44) | 76.20 (20) | 75.70 (38) | 76.30 (32) | 78.76 (28) | 0 (0) |
| IDInit | 71.60 (42) | 76.71 (14) | 76.72 (24) | 76.93 (22) | 79.10 (23) | 0.55 (7.4) |

convergence and better performance than the baseline. And when training RN-50 with Adamw, the convergence of IDInit is still consistently fast. Compared with RN-50, our initialization shows a faster convergence speed. IDInit has an average improvement of 0.55%, which is significant to be in practice. This experiment shows the good practicability and promising probability of IDInit, which is beneficial to the artificial intelligence community.

### 4.4 ABLATION EXPERIMENT

We conduct this experiment to validate the effect of the proposed two improvements. The dataset is Cifar10 and the backbone is ResNet-20. We run four times following settings: (i) IDInit w/o $IDIC_\tau$ and w/o $IDIZC_\varepsilon$; (ii) IDInit w/o $IDIC_\tau$ and w/ $IDIZC_\varepsilon$; (iii) IDInit w/ $IDIC_\tau$ and w/o $IDIZC_\varepsilon$; (iv) IDInit. We choose SGD with momentum 0.9, weight decay 5e-4 and learning rate 0.1 to train the models for 200 epochs. The learning rate is reduced with a cosine function.

As shown in Table 4, by applying the identity matrix directly, (i) obtains the lowest accuracy of 87.01% among all cases. Regarding results

Table 4: Results of the ablation experiment on ResNet-20.

| Setting | (i) | (ii) | (iii) | (iv) |
|---|---|---|---|---|
| Accracy | $87.01_{\pm 0.29}$ | $92.9_{\pm 0.18}$ | $90.43_{\pm 0.14}$ | $\mathbf{93.22}_{\pm 0.05}$ |

of (ii) and (iii), both the two settings can make significant improvements of nearly 5.89% and 3.42% from (i), respectively. And $IDIZC_\varepsilon$ can make a deeper effect than $IDIC_\tau$. Equipping $IDIC_\tau$ and $IDIZC_\varepsilon$, IDInit will improve performance further, which demonstrates our modification is efficient.

### 4.5 TEXT CLASSIFICATION

We implement text classification on SST2 (Socher et al., 2013) and TREC-6 (Li & Roth, 2002) and select TextCNN (Kim, 2014), TextRNN (Lai et al., 2015) and Transformer (Vaswani et al., 2017) for comparison. For TextCNN and TextRNN, we use AdaDelta (Zeiler, 2012) optimizer with a learning

Table 5: Results of TextCNN and TextRNN on SST2 and TREC-6. The subscript G denotes the embedding layer is initialized by Glove while W indicates Word2Vec. "Default" means the default initialization of models, specifically, Kaiming for TextCNN, and Xavier for both TextRNN and Transformer. Std values larger than 1.0 are marked in red. More results can be found in Table 7.

| Datasets | Init. | TextCNN$_{G/W}$ | TextRNN$_{G/W}$ | Transformer$_{G/W}$ |
|---|---|---|---|---|
| SST2 | Default | $81.40_{\pm0.66}/84.56_{\pm0.43}$ | $81.69_{\pm0.30}/84.29_{\pm0.70}$ | $80.97_{\pm1.20}/83.36_{\pm0.76}$ |
| | Orthogonal | $82.24_{\pm0.44}/84.37_{\pm0.38}$ | $81.86_{\pm0.55}/84.61_{\pm0.78}$ | $82.22_{\pm0.87}/83.99_{\pm0.23}$ |
| | IDInit | $\mathbf{82.60}_{\pm0.24}/\mathbf{85.67}_{\pm0.41}$ | $\mathbf{82.66}_{\pm0.16}/\mathbf{85.49}_{\pm0.33}$ | $\mathbf{82.48}_{\pm0.55}/\mathbf{84.51}_{\pm0.24}$ |
| TREC-6 | Default | $90.80_{\pm0.94}/92.06_{\pm1.00}$ | $86.34_{\pm1.04}/90.52_{\pm1.54}$ | $86.68_{\pm2.68}/89.20_{\pm1.20}$ |
| | Orthogonal | $90.34_{\pm0.72}/92.72_{\pm0.84}$ | $85.86_{\pm0.90}/89.88_{\pm1.54}$ | $86.90_{\pm1.51}/89.26_{\pm0.86}$ |
| | IDInit | $\mathbf{91.22}_{\pm0.54}/\mathbf{92.94}_{\pm0.48}$ | $\mathbf{87.04}_{\pm0.26}/\mathbf{90.60}_{\pm0.58}$ | $\mathbf{87.32}_{\pm0.78}/\mathbf{90.06}_{\pm0.60}$ |

rate 1.0 and adopt Adam (Kingma & Ba, 2015) for Transformer with a learning rate 1e-4. For the embedding layer, we utilize Glove (Pennington et al., 2014) and Word2Vec(Mikolov et al., 2013) to initialize the embedding weights. All models are trained up to 10 epochs for 5 times.

As shown in Table 5, all the initialization methods can work normally. Default random initialization obtains the lowest accuracy in most cases on both SST2 and TREC-6. Orthogonal initialization always derives modest results. By contrast to baselines, IDInit can achieve the highest accuracy in all conditions. In addition, IDInit always obtains the smallest std values, showing stable performance.

## 4.6 Pre-Training on Language Model

Pre-training plays an important role in various applications. We conduct the experiment to show the fast convergence on BERT (Devlin et al., 2019). The dataset is the concatenation of English Wikipedia and Toronto Book Corpus Zhu et al. (2015). We train the BERT-Base for 40 epochs with 768 batch size. The optimizer is set to AdamW with learning rate 1e-4 and weight decay 1e-2. 32 NVIDIA V100s are used.

As shown in Figure 9, "Default" means the default initialization of models for BERT-Base. IDInit achieves faster convergence. Specifically, IDInit shows an 11.3% acceleration ratio in terms of FLOPs. Moreover, IDInit can derive a lower loss of 1.46 in the end. As a result, IDInit is promising used in practice for enhancing convergence ability and performance.

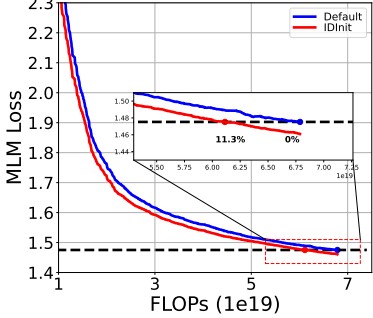

Figure 9: Results of BERT-Base.

## 5 Conclusion

An efficient initialization approach is crucial for training deep neural networks. In this paper, we introduce a fully identical initialization (IDInit) that is based on the identity matrix. Addressing the problems encountered when developing IDInit, i.e., dead neurons and performance degeneration, we give two concise solutions, namely using small numerical values to wipe off dead neurons and reshaping an identity-like matrix into a tensor thus increasing feature diversity, leading to a performance improvement. With good performance on wide generality, high stability, and fast convergence, IDInit is promising to be applicable in practice. In the future, we hope that this identical design can motivate the AI community to implement more novel initialization methods.

**Limitation.** We defer the limitation of the proposed IDInit in Appendix D.

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

# A  IDInit Details

## A.1  Full IDInit Scheme

Here, we show the full IDInit scheme in Figure 10.

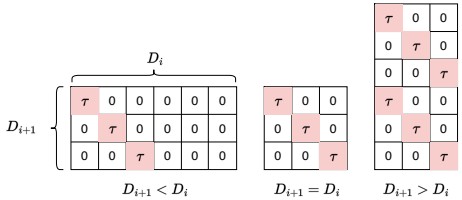

Identity Preserving Initialization

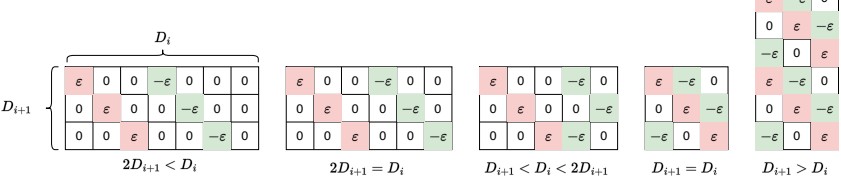

Zero Preserving Initialization

Figure 10: Illustration of IDInit with all conditions.

## A.2  Trainability Analysis of Non-Squared Matrices.

The magnitude of signals transiting in layers is usually related to stability, and the variance of signals can be a good indicator of the magnitude according to the variance-control mechanism. Obviously, $IDI_\tau$ can maintain forward signal variance. Therefore, we explore the trainability from the angle of the gradient backward procedure. Prior to that, consider a network $x^{(L)} = f(x|\Theta) = \theta^{(L-1)}\theta^{(L-2)}\dots\theta^{(0)}x^{(0)}$, the gradient relationship can be formulated as

$$\frac{\partial \mathcal{L}}{\partial x^{(i)}} = \theta^{(i)\mathrm{T}}\frac{\partial \mathcal{L}}{\partial x^{(i+1)}} = \theta^{(i)\mathrm{T}}\theta^{(i+1)\mathrm{T}}\dots\theta^{(L-1)\mathrm{T}}\frac{\partial \mathcal{L}}{\partial x^{(L)}}, \qquad \frac{\partial \mathcal{L}}{\partial \theta^{(i)}} = \frac{\partial \mathcal{L}}{\partial x^{(i+1)}}x^{(i)\mathrm{T}}, \qquad (9)$$

where $x^{(i)} \in \mathbb{R}^{D_i}$, $\theta^{(i)} \in \mathbb{R}^{D_{i+1}\times D_i}$ and $\mathcal{L}$ denotes a loss of training. Since $\forall i, D_{i+1} = D_i$, the training process will be stable as that gradients transit identically. Here, we pay attention to non-square conditions. $\tau = 1$ for there is no activation function.

If $\forall i, D_{i+1} < D_i$, then $\frac{\partial \mathcal{L}}{\partial x^{(i)}}$ ($i \in \{1, 2, \dots, L-1\}$) contains elements of 0, which indicates that $\frac{\partial \mathcal{L}}{\partial \theta^{(i)}}$ must contain 0 values, causing failed updating. However, this is a fake dead phenomenon, since all weights will be trained after several updating steps. At the first training step, $\frac{\partial \mathcal{L}}{\partial \theta^{(L-1)}} = \frac{\partial \mathcal{L}}{\partial x^{(L)}}x^{(L-1)\mathrm{T}}$. For both $\frac{\partial \mathcal{L}}{\partial x^{(L)}}$ and $x^{(L-1)}$ not definitely containing 0 values, $\theta^{(L-1)}$ can be updated normally and deviates from identity-like form. Therefore, $\frac{\partial \mathcal{L}}{\partial x^{(L-1)}}$ can get normal updating in the next step. As a result, all weights can be updated normally.

If $\forall i, D_{i+1} > D_i$, then the variance of gradients will increase from layer to layer. However, this will not make trouble for training. For $\frac{\partial \mathcal{L}}{\partial x^{(i)}}$, its variance $\sigma^2(\frac{\partial \mathcal{L}}{\partial x^{(i)}}) \approx \frac{D_{i+1}}{D_i}\sigma^2(\frac{\partial \mathcal{L}}{\partial x^{(i+1)}})$. Therefore, the variance of $\frac{\partial \mathcal{L}}{\partial x^{(0)}}$ can be calculated as

$$\sigma^2(\frac{\partial \mathcal{L}}{\partial x^{(0)}}) = \frac{D_1}{D_0}\frac{D_2}{D_1}\dots\frac{D_L}{D_{L-1}}\sigma^2(\frac{\partial \mathcal{L}}{\partial x^{(L)}}) = \frac{D_L}{D_0}\sigma^2(\frac{\partial \mathcal{L}}{\partial x^{(L)}}). \qquad (10)$$

According to Eq. (10), the gradient will not explode in the backward procedure.

### A.3 PROOF FOR THEOREM 3.1.

*Proof.* Assume weights are updated with the stochastic gradient descent (SGD). Without loss of generality, we set $D_h = 2D_0 = 2D_L$. Given two batches of inputs as $\{x_1^{(0,1)}, x_1^{(0,2)}, \ldots, x_1^{(0,N)}\} \in \mathbb{R}^{D_0}$ and $\{x_2^{(0,1)}, x_2^{(0,2)}, \ldots, x_2^{(0,N)}\} \in \mathbb{R}^{D_0}$, where $N \geq D_0$ is the batch size. Therefore, the initial gradients of weights are

$$\frac{\partial \mathcal{L}}{\partial \theta^{(0)}} = \begin{pmatrix} \Pi \\ \mathbf{0} \end{pmatrix}, \qquad \frac{\partial \mathcal{L}}{\partial \theta^{(1)}} = \begin{pmatrix} \Pi & \Pi \\ \mathbf{0} & \mathbf{0} \end{pmatrix}, \qquad \frac{\partial \mathcal{L}}{\partial \theta^{(2)}} = \begin{pmatrix} \Pi & \Pi \end{pmatrix},$$

where $\frac{\partial \mathcal{L}}{\partial \theta^{(0)}} \in \mathbb{R}^{D_h \times D_0}$, $\frac{\partial \mathcal{L}}{\partial \theta^{(1)}} \in \mathbb{R}^{D_h \times D_h}$, $\frac{\partial \mathcal{L}}{\partial \theta^{(2)}} \in \mathbb{R}^{D_L \times D_h}$, $\Pi = \frac{1}{N} \sum_{i=1}^{N} \frac{\partial L}{\partial x^{(L)}} \circ x_1^{(0,i)} \in \mathbb{R}^{D_L \times D_0}$, and $\mathbf{0}$ is a zero values. $\circ$ denotes outer production.

After training with the second data batch, the gradient is calculated as follows:

$$\frac{\partial \mathcal{L}}{\partial \theta^{(1)}} = \begin{pmatrix} (I - \mu\Pi)M(I - \mu\Pi)K & (I - \mu\Pi)MK \\ -\mu\Pi M(I - \mu\Pi)K & -\mu\Pi MK \end{pmatrix}, \tag{11}$$

where $M = \frac{\partial \mathcal{L}}{\partial x_2^{(3)}}$ and $K = \frac{1}{N} \sum_{i=1}^{N} x_2^{(0,i)}$. This leads to the following residual component:

$$\hat{\theta}^{(1)} = \theta^{(1)} - I = \begin{pmatrix} -\mu\Pi - \mu(I - \mu\Pi)M(I - \mu\Pi)K & -\mu\Pi - \mu(I - \mu\Pi)MK \\ \mu^2\Pi M(I - \mu\Pi)K & \mu^2\Pi MK \end{pmatrix}, \tag{12}$$

Without loss of generality, assuming $rank(\Pi) = D_0$, we can conclude

$$rank(\hat{\theta}^{(1)}) \geq D_0. \tag{13}$$

Therefore, IDInit can break the rank constraint by achieving the rank of $\hat{\theta}^{(1)}$ larger than $D_0$.

$\square$

### A.4 IMPLEMENTING IDINIT ON ATTENTION LAYER IN TRANSFORMER

In this part, we show the way to initialize the attention layer with IDInit. Prior to that, formulating an attention layer as

$$\text{Att}(Q, K, V) = \text{softmax}(\frac{QW^Q W^K K}{\sqrt{d}})VW^V W^O, \tag{14}$$

where $Q$ is the query matrix, $K$ means the key matrix, $V$ denotes the value matrix, $W^Q$, $W^K$ and $W^V$ represents the weights for $Q$, $K$, and $V$ respectively, and $W^O$ is the output transformation. Following the instruction of IDInit in Sec. 3, we firstly use $\text{IDI}_\tau$ to initialize $W^Q$, $W^K$, $W^V$ and $W^O$. And then, we use $\text{IDIZ}_\varepsilon$ to initialize the last fully-connected layer $W^O$. The $\tau$ and $\varepsilon$ are consistently set with the paper content to 1 and 1e-6, respectively.

### A.5 DETAILS OF PATCH-MAINTAIN CONVOLUTION

We illustrate the figure to show the comparison between channel-maintain convolution and patch-maintain convolution in Figure 11.

## B DETAILED SETTINGS OF EXPERIMENTS

In this paper, for ReLU activated networks, $\tau$ is set to $\sqrt{2}$ for the first layer in a network and 1 for other $\text{IDI}_\tau$ / $\text{IDIC}_\tau$ initializing layers, while for tanh-activated networks, all $\text{IDI}_\tau$ is set to 1, and $\varepsilon$ is $1e-6$ for all $\text{IDIZ}_\varepsilon$ / $\text{IDIZC}_\varepsilon$ initializing layers.

### B.1 DETAILS OF VALIDATION ON NON-RESIDUAL CONVOLUTION EXPERIMENT

In this experiment, we use AllConv (Springenberg et al., 2015) which consists of nine convolutional layers as the backbone network. We show the structure of AllConv in Table 6. The dataset is Cifar10.

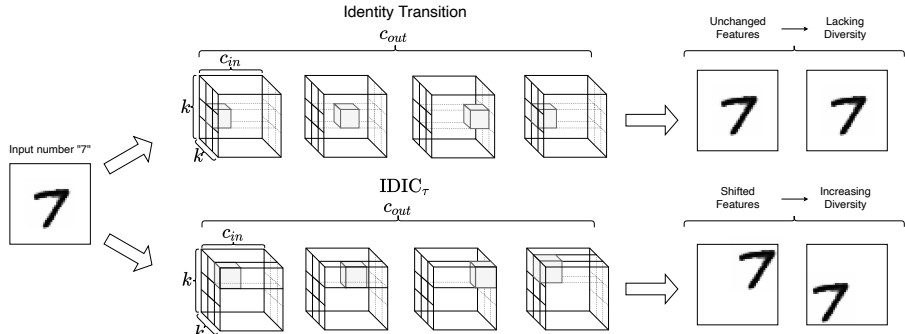

Figure 11: A case of number "7" on Identical Convolution Layer. The upper sub-figure maintains the identity transition. The under sub-figure is IDIC$_\tau$ initialization that shifts features for increasing diversity. More feature diversity from IDIC$_\tau$ is beneficial for improving model performance.

The optimizer is Stochastic Gradient Descent (SGD) with momentum 0.9, weight decay 5e-4, and learning rate 1e-1. The learning rate scheduler adopts a warm-up cosine reduction strategy. We run the model in 300 epochs on one Nvidia A100. We adopt Kaiming initialization and IDInit w/o IDIC$_\tau$ initialization for comparison. Since there is no residual connection, we do not consider the IDIZC$_\varepsilon$ function in this experiment. For each initialization, we have run them with 0, 10, 20, 30, 40, 50, and 60 warm-up epochs. The experiment is conducted on one Nvidia A100.

Table 6: Architectures of the tensorial All-Conv networks. Window means the convolutional kernel window size. Channels indicate $\mathbf{c}_{in}$ and $\mathbf{c}_{out}$ of a standard convolutional kernel $\mathcal{C} \in \mathbb{R}^{\mathbf{c}_{in} \times \mathbf{c}_{out} \times k \times k}$. The avg pool denotes the average pooling operation.

| Layer | Window | Channels |
|-------|--------|----------|
| conv1 | 3×3 | 3× 96 |
| conv2 | 3×3 | 96× 96 |
| conv3 | 3×3 | 96× 96 |
| conv4 | 3×3 | 96× 192 |
| conv5 | 3×3 | 192× 192 |
| conv6 | 3×3 | 192× 192 |
| conv7 | 3×3 | 192× 192 |
| conv8 | 1×1 | 192× 192 |
| conv9 | 1×1 | 192× 10 avg pool |

## B.2 DETAILS OF VALIDATION ON RESIDUAL CONVOLUTION EXPERIMENT

In this experiment, we validate the proposed initialization with the comparison with existing initialization, including (1) Fixup; (2) SkipInit; (3) ReZero; (4) Kaiming; (5) Zero $\gamma$ (Setting the scale in Batch Normalization (BN) to 0). We use ResNet-56/110 as backbones on Cifar10. To verify the universality, we use two settings, namely w/ and w/o BN. For analyzing convergence, we adopt both SGD and Adam optimizer for updating models. We set SGD, with the momentum 0.9, the weight decay 5e-4, and the learning rate 0.2. For Adam, the learning rate is 0.001, $\beta_1$ is 0.9 and $\beta_2$ is 0.999. We train models for 200 epochs. The learning rate is reduced with a cosine function. The experiment is conducted on one Nvidia A100.

## B.3 DETAILS OF ABLATION EXPERIMENT

The dataset is Cifar10 and the backbone is ResNet-20. We choose SGD with momentum 0.9, weight decay 5e-4, and learning rate 0.1 to train the models for 200 epochs. The learning rate is reduced

with a cosine function. And data-augment mixup is applied. The experiment is conducted on one Nvidia A100.

## B.4 DETAILS OF TEXT CLASSIFICATION EXPERIMENT

We also explore performance networks on text classification datasets including SST2, SST5 (Socher et al., 2013) and TREC-6, and we select TextCNN (Kim, 2014), TextRNN (Lai et al., 2015) and Transformer (Vaswani et al., 2017) for comparison. For TextCNN and TextRNN, we use AdaDelta (Zeiler, 2012) optimizer with a learning rate 1.0 and adopt Adam (Kingma & Ba, 2015) for Transformer with a learning rate 1e-4. For the embedding layer, we utilize Glove (Pennington et al., 2014) and Word2Vec (Mikolov et al., 2013) to initialize the embedding weights. All models are trained up to 10 epochs, and we run all the random initialization 5 times. The experiment is conducted on one Nvidia A100.

Table 7: Results of TextCNN and TextRNN on SST2, SST5 and TREC-6. The subscript G denotes the embedding layer is initialized by Glove while W indicates Word2Vec. "Default" means the default initialization of models, specifically, Kaiming for TextCNN, and Xavier for both TextRNN and Transformer. Std values larger than 1.0 are marked in red.

| Datasets | Init. | TextCNN$_{G/W}$ | TextRNN$_{G/W}$ | Transformer$_{G/W}$ |
|---|---|---|---|---|
| SST2 | Default | $81.40_{\pm0.66}/84.56_{\pm0.43}$ | $81.69_{\pm0.30}/84.29_{\pm0.70}$ | $80.97_{\pm1.20}/83.36_{\pm0.76}$ |
| | Orthogonal | $82.24_{\pm0.44}/84.37_{\pm0.38}$ | $81.86_{\pm0.55}/84.61_{\pm0.78}$ | $82.22_{\pm0.87}/83.99_{\pm0.23}$ |
| | IDInit | $\mathbf{82.60}_{\pm0.24}/\mathbf{85.67}_{\pm0.41}$ | $\mathbf{82.66}_{\pm0.16}/\mathbf{85.49}_{\pm0.33}$ | $\mathbf{82.48}_{\pm0.55}/\mathbf{84.51}_{\pm0.24}$ |
| SST5 | Default | $44.68_{\pm0.88}/46.15_{\pm0.62}$ | $44.27_{\pm0.88}/47.04_{\pm0.48}$ | $41.81_{\pm1.17}/44.02_{\pm1.27}$ |
| | Orthogonal | $44.91_{\pm0.81}/46.76_{\pm0.68}$ | $44.61_{\pm1.18}/46.13_{\pm0.79}$ | $43.01_{\pm1.61}/44.92_{\pm1.52}$ |
| | IDInit | $\mathbf{45.41}_{\pm0.40}/\mathbf{47.72}_{\pm0.51}$ | $\mathbf{46.65}_{\pm0.25}/\mathbf{48.01}_{\pm0.36}$ | $\mathbf{44.23}_{\pm0.59}/\mathbf{45.76}_{\pm0.42}$ |
| TREC-6 | Default | $90.80_{\pm0.94}/92.06_{\pm1.00}$ | $86.34_{\pm1.04}/90.52_{\pm1.54}$ | $86.68_{\pm2.68}/89.20_{\pm1.20}$ |
| | Orthogonal | $90.34_{\pm0.72}/92.72_{\pm0.84}$ | $85.86_{\pm0.90}/89.88_{\pm1.54}$ | $86.90_{\pm1.51}/89.26_{\pm0.86}$ |
| | IDInit | $\mathbf{91.22}_{\pm0.54}/\mathbf{92.94}_{\pm0.48}$ | $\mathbf{87.04}_{\pm0.26}/\mathbf{90.60}_{\pm0.58}$ | $\mathbf{87.32}_{\pm0.78}/\mathbf{90.06}_{\pm0.60}$ |

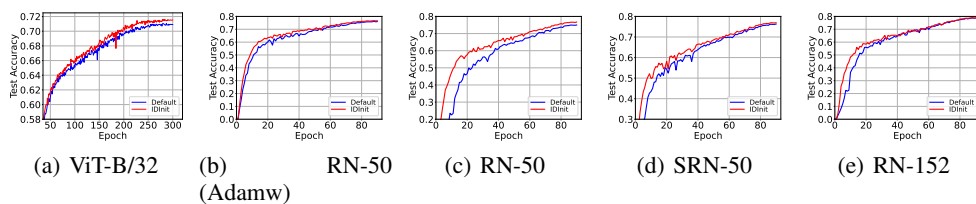

(a) ViT-B/32    (b)    RN-50    (c) RN-50    (d) SRN-50    (e) RN-152
(Adamw)

Figure 12: Results on ImageNet. "Default" means the default initialization of models. RN-50 (Adamw) means that ResNet-50 is trained with the same optimizer Adamw as the ViT-B/32.

## B.5 DETAILS OF IMAGE CLASSIFICATION ON IMAGENET EXPERIMENT

In this experiment, we use ImageNet for validation. We use ViT-B/32 (Dosovitskiy et al., 2021), ResNet-50/152 (RN-50/152) and Se-ResNet-50 (SRN-50) as backbones. For ViT-B/32 that inputs $32 \times 32$ patch window, the optimizer is AdamW with a learning rate 1e-3 and a weight decay of 5e-2. And the batch size is 1024. The epoch for training is 300. We use 30 epochs for warm-up. The input image size is $224 \times 224$. The dropout rates of the embedding layer and the network layer are all 0.1. For RN-50/152 and SRN-50, the optimizer is SGD with a learning rate 1e-1 and a weight decay of 1e-4. And the batch size is 1024. The epoch for training is 90. We use 9 epochs for warm-up. The

input image size is $160 \times 160$ for the front 35 epochs and $224 \times 224$ for the remaining epochs. For all models, we apply data-augment including cutmix (Yun et al., 2019) with $\alpha = 1.0$, mixup (Zhang et al., 2018) with $\alpha = 0.8$, the switching probability is 0.5 and a label smoothing with 0.1. The experiment is conducted on 4 Nvidia A100.

### B.6 DETAILS OF PRE-TRAINING ON LANGUAGE MODEL

Pre-training plays an important role in various applications. We conduct the experiment to show the fast convergence on BERT (Devlin et al., 2019). The dataset is the concatenation of English Wikipedia and Toronto Book Corpus Zhu et al. (2015). We train the BERT-Base for 40 epochs with 768 batch size. The optimizer is set to AdamW with learning rate 1e-4 and weight decay 1e-2. 32 NVIDIA V100s are used.

Table 8: Results of Linear-5 on MNIST. "Default" means the default initialization of models where Xavier is for Linear-5-tanh and Kaiming is adopted for Linear-5-ReLU.

| Init. | Linear-5-tanh | Linear-5-ReLU |
|---|---|---|
| Default | 98.26 | 98.21 |
| IDInit | **98.32** | **98.4** |

## C ADDITIONAL EXPERIMENTS

We provide additional experiments to further validate IDInit. $\tau$ and $\varepsilon$ are set the same as Sec. B.

### C.1 VALIDATION ON THE LINEAR STRUCTURE

This experiment is conducted on MNIST. We use five linear layers named Liner-5 whose hidden layers are all of dimension 512. The optimizer is SGD with momentum 0.9, weight decay 5e-4, and a learning rate 1e-1. The learning rate scheduler adopts a cosine reduction strategy. We run the model in 30 epochs on one Nvidia A100. We both consider Linear-5-tanh and Linear-5-ReLU which consist of Linear-5, and tanh and ReLU activation functions, respectively. The experiment is conducted on one Nvidia A100.

As shown in Table 8, IDInit can achieve the highest accuracy in both different tanh and ReLU conditions. The results show the ability of our proposed method to train a model with only fully-connected layers.

### C.2 ANALYSIS ON VARIANCE PROPAGATION

Here we conduct an experiment on Cifar10 to demonstrate data-flow will keep stable. We use 4 types of networks: (1) FC: 10-layer fully-connected layers; (2) ResFC: 10 residual blocks (two fully-connected layers in a block); (3) Conv: 9-layer AllConv in Sec. B.1; (4) ResConv: 10 residual blocks (two convolutional layers in a block). For (1) and (2) two fully-connected networks, we reshape Cifar10 data as $\mathbf{X} \in \mathbb{R}^{32 \times 96}$ as input and does not use any activation function. For (1), hidden lengths are $\{200, 400, 600, 800, 1000, 1000, 800, 600, 400, 200\}$. For (2), hidden lengths are all set to 96. For (3) and (4) two convolution networks, we directly input images to them, and use ReLU as the activation function. For (3), we directly use AllConv as shown in Table 6. For (4), we first use convolution to transfer an image to 16 channels, and then set the channels of all convolution within residual blocks to 16. For comparison, we use Xavier for (1) and (2), and Kaiming for (3) and (4) in terms of the activation function. We also employ noises with 0 mean, and $\{0.00, 0.01, 0.10, 1.00\}$ for comparing robustness. In the experiment, we run 500 rounds for each model. The experiment is conducted on one Nvidia A100.

Results are shown in Figure 13. The regular methods Xavier and Kaiming can only work on non-residual networks. On residual networks, they both cause giant standard derivation, leading to in-

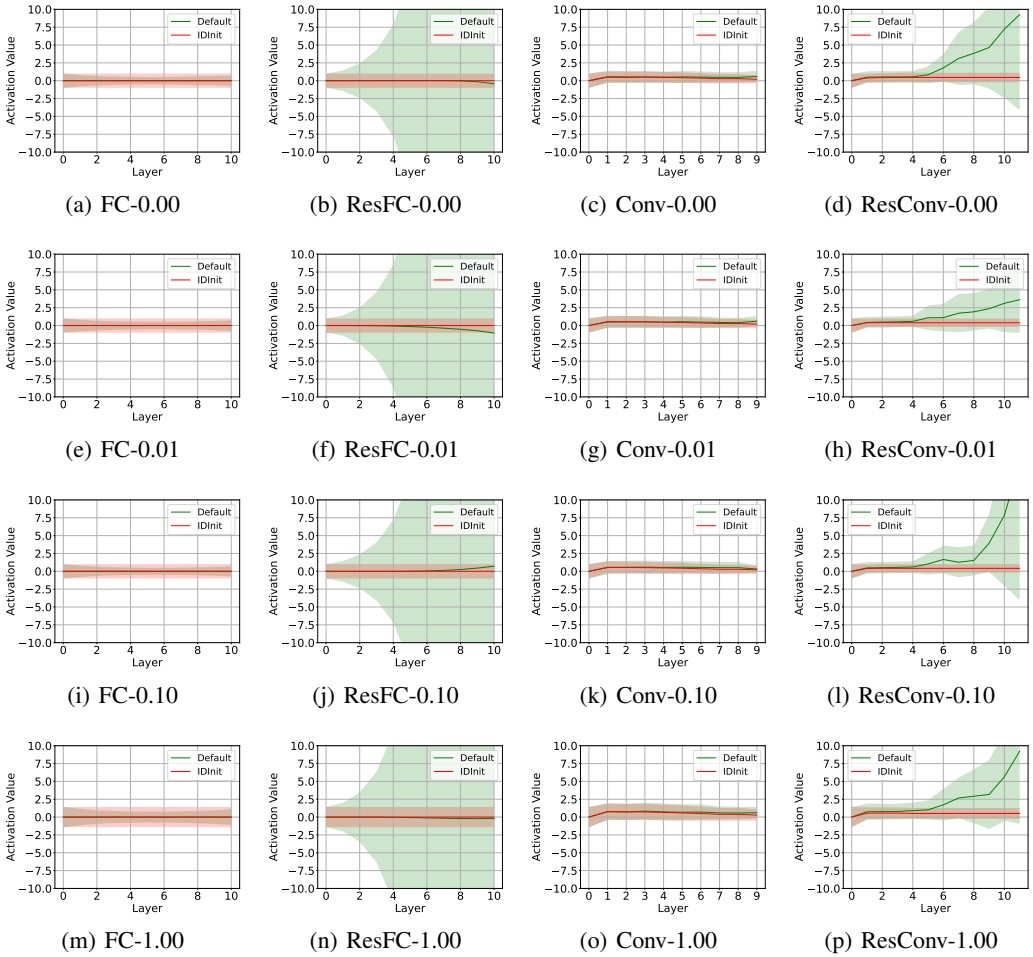

Figure 13: Results of the analysis on variance propagation. The numerical value after the model name means the standard derivation of the noise. "Default" means the default initialization of models, specifically, Xavier for FC and ResFC, and Kaiming for Conv and ResConv. The default methods can only work on non-residual networks FC and Conv, however, fail on residual networks ResFc and ResConv, for cause instability with giant standard derivation. By contrast, IDInit can consistently transit data-flow in an appropriate scale on all models and various noises, which shows sufficient robustness, and can provide models with stable and efficient training.

stability. By contrast, the proposed IDInit can consistently transit data-flow in an appropriate scale on all models and various noises, which shows sufficient robustness, and can provide models with stable and efficient training.

## C.3 ANALYSIS ON WEIGHT DISTRIBUTION

In this experiment, we conduct an experiment on Cifar10 with ResNet-20 to show the weight distribution of IDInit. We use an SGD optimizer with a learning rate 0.2, and weight decay 5e-4. The batch size is 1024. Training epochs are 200. The learning rate is reduced with a cosine function. The experiment is conducted on one Nvidia A100.

The results are shown in Figure 14, weights initialized with IDInit are almost full of zero at the beginning, while Kaiming uses a Gaussian distribution. At the end of the training, IDInit still contains more zero values than Kaiming, which is beneficial for memory occupation since a 0 value will not cost memory space.

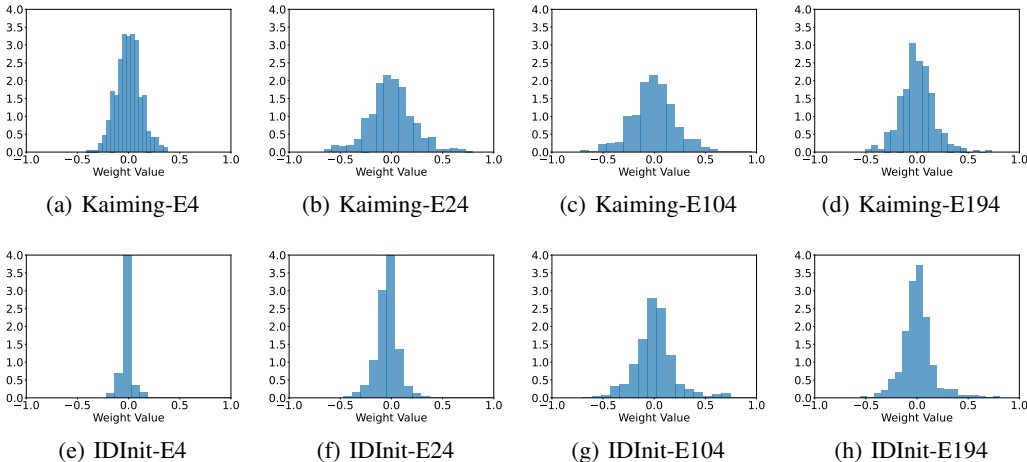

Figure 14: Histograms of the first convolution weights in ResNet-20. "E" means the epoch index. IDInit contains more zero values in each epoch compared with Kaiming initialization.

## C.4 ANALYSIS ON INPUT-OUTPUT JACOBIAN

Here we conduct an experiment on Cifar10 with 64 blocks in Figure 1 to demonstrate IDInit follows the dynamical isometry. We remove batch normalization for the more clear difference between IDInit and Kaiming. We use an Adagrad optimizer with a learning rate 0.01. The batch size is 100. The activation is ReLU. The experiment is conducted on one Nvidia A100.

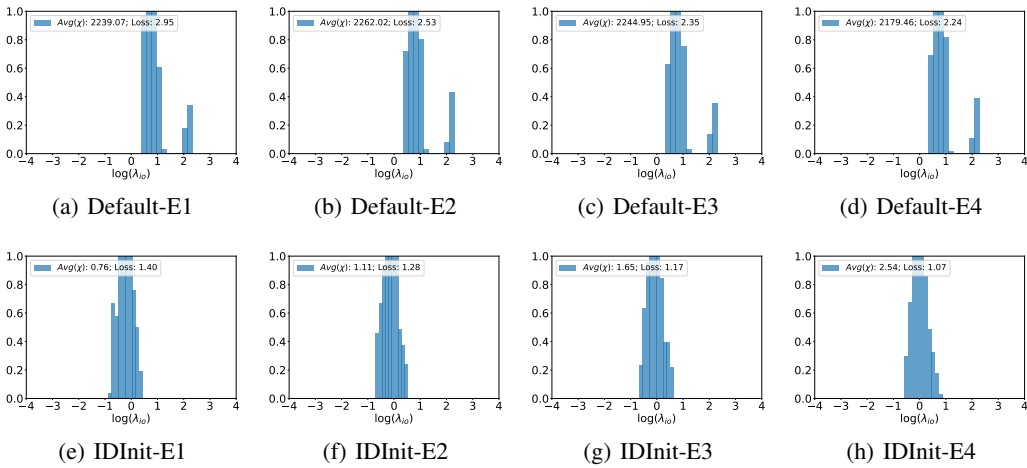

Figure 15: Histograms of log singular values ($\log(\lambda_{io})$) for the input-output Jacobian. "E" means the epoch index. Compared with Default initialization, IDInit has a significantly smaller squared singular value $\chi$, which can achieve a faster reduction of the loss.

As shown in Figure 15, Default initialization cause a high squared singular value $\chi$, reaching more than 2000. Compared to Default, IDInit only derives $\chi$ around 1, indicating correspondence to the dynamical isometry. In addition, the loss of IDInit decreases faster than Default, which shows a good convergent ability.

## C.5 FAILURE OF LONG RESIDUAL STEM

We conduct this experiment to show the failure case when the residual stem is long to show the importance of the stability of the residual stem. In this experiment, we conduct an experiment on

Cifar10. We use a residual network named Res-112 as in Table 9. We set 109 layers in the residual stem. Batch normalization is not applied for fairly validating the stability of initialization methods. We use an SGD optimizer with a learning rate 0.2, and weight decay 1e-8. The batch size is 768. Training epochs are 35. The learning rate is reduced with a cosine function. One Nvidia A100 is used.

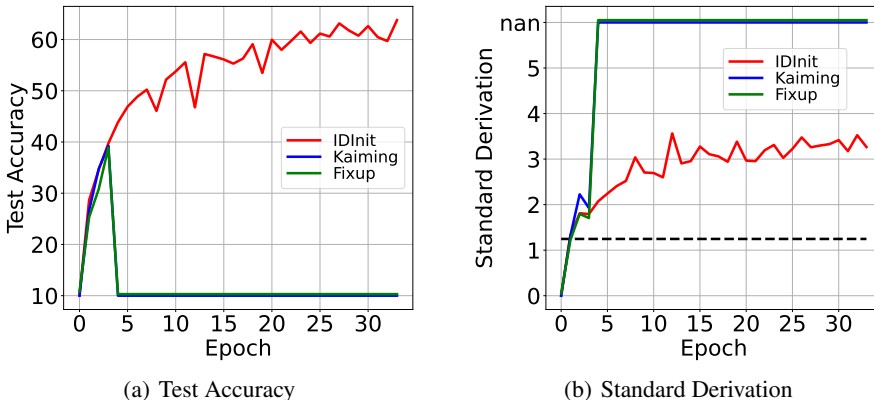

(a) Test Accuracy        (b) Standard Derivation

Figure 16: Result of the experiment on the residual network with the long residual stem. Figure 16(a) shows the accuracy of different initialization. Figure 18(c) shows the standard derivations of the outputs of networks with different initialization methods. The black dash line is the standard derivation of the network input.

Results are shown in Figure 16. When the network is trained for 4 epochs, both Kaiming and Fixup fail to train the network, since the standard derivations of their outputs explode. By contrast, IDInit successfully trains this network and the standard derivation of the output converges to a stable value. This experiment demonstrates the ability of IDInit to stabilize the residual stem, which can benefit the training of the whole network.

Table 9: Architectures of Res-112. Window means the convolutional kernel window size. Channels indicate $\mathbf{c}_{in}$ and $\mathbf{c}_{out}$ of a standard convolutional kernel $\mathcal{C} \in \mathbb{R}^{\mathbf{c}_{in} \times \mathbf{c}_{out} \times k \times k}$. The avg pool denotes the average pooling operation. Linear means a linear layer.

| Layer | Window | Channels |
|---|---|---|
| conv1 | 3×3 | 3×16 |
| Residual Block | 3×3 | [16×16]×18 |
| | 3×3 | 16× 32 |
| | | [32×32]×17 |
| | 3×3 | 32× 64 |
| | | [64×64]×17 |
| | 3×3 | 64×64 |
| conv2 | 3×3 | 64×64 |
| | | avg pool |
| Linear | | 64×10 |

# D  LIMITATION

While IDInit demonstrates notable advancements in convergence speed and performance enhancement, it still possesses certain limitations.

(1) Networks initialized with identity matrices face challenges in converging to ground truths that include negative eigenvalues. However, this drawback can be easily mitigated by incorporating

momentum into the optimizer. Given that momentum is a commonly used setting, this limitation can be implicitly resolved as we show in the main context.

(2) The deterministic nature of IDInit may pose difficulties in addressing concerns regarding model properties solely through model selection. Nevertheless, even with limited variation caused by determinacy when the seed changes, IDInit consistently exhibits excellent performance across different settings, including various seed selections. This suggests that random initialization might not surpass IDInit, even with carefully chosen seeds. Additionally, the randomness introduced by the sequence order of data batches during training also contributes to the overall randomness and ultimately influences the model's final performance. Hence, it seems unlikely that IDInit's determinacy significantly impacts model selection. Moreover, determinacy can reduce training variation, resulting in improved training reproducibility, which cannot be achieved by random initialization.

Hence, these two limitations are not expected to cause severe harm to IDInit.

# E    DYNAMICAL ISOMETRY IN IDINIT

Following Bachlechner et al. (2021), we utilize a simple example of the mechanism that dynamical isometry helps IDInit to obtain a fast convergence. Considering a $L$-layer network with a simple special case of Eq. (1):

$$x^{(L)} = (r + w^{(2)}w^{(1)})^L x^{(0)}, \qquad (15)$$

where $w^{(1)}$ and $w^{(2)}$ denote the first weight and last weight in a residual stem respectively, and $x^{(*)}$ is the feature in layers. $r \in \{0,1\}$ determines residual connection. Specifically, $r = 0$ and $r = 1$ represent non-residual and residual conditions respectively. The Jacobian of Eq. (15) is $J_{0L} = (r + w^{(2)}w^{(1)})^L$. Obviously, identity transition on both non-residual and residual settings, namely $\{r = 0, w^{(2)} = w^{(1)} = 1\}$ and $\{r = 1, w^{(1)} = 1, w^{(2)} = 0\}$ respectively, will achieve $J_{0L} = 1$, which conforms to the dynamical isometry mechanism that helps improving training ability (Pennington et al., 2017). Further, we delve into a gradient update analysis. Following gradient descent, $w_1$ can be updated with

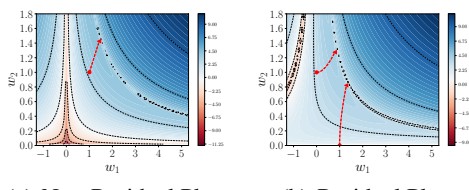

(a) Non-Residual Plot.   (b) Residual Plot.

Figure 17: Contour plots of the log gradient norm $\log \|\partial R\|_2$ on non-residual and residual networks. $w^{(1)}$ and $w^{(2)}$ are both weights. The training process set as Bachlechner et al. (2021), which is conducted on ground-truth $x^{(L)} = 50 \times x_0$ via gradient descent using a training set of $x_0 = \{1., 1.1, ..., 1.8\}$. (a) shows $\{w^{(2)} = w^{(1)} = 1\}$ can avoid poorly conditioned regions around 0, and converge to $w^{(1)}w^{(2)} = 2.19$. (b) cares about two initial position $\{w^{(1)} = 0, w^{(2)} = 1\}$ and $\{w^{(2)} = 1, w^{(1)} = 0\}$. The two points' trajectories do not also pass the poor regions around $w^{(1)} = -1, w^{(2)} = 1$ and converge to the solution $w^{(1)}w^{(2)} = 1.19$.

$$\Delta w^{(1)} = -\lambda L w^{(2)} x^{(0)} (r + w^{(2)}w^{(1)})^{L-1} \partial_x R(x)|_{x=x^{(L)}}, \qquad (16)$$

where $R$ means the loss function, and $\lambda$ is a learning rate. As $w^{(1)}$ and $w^{(2)}$ are equivalent in Eq. (15), $w^{(2)}$ can be updated similar to Eq. (16). When $w^{(1)} = 1$, updates are required less than 1. Therefore, the learning rate is constrained to

$$\begin{cases} \lambda \propto L^{-1}, & \text{if non-residual,} \\ \lambda \propto L^{-1}(1 + w^{(2)})^{L-1}, & \text{if residual.} \end{cases} \qquad (17)$$

For the non-residual condition, the learning rate is polynomial to $L$, thereby insensitive to the depth. By contrast, in the residual block, $w^{(2)} >> 0$ will cause learning rate exponentially small and $w^{(2)} = -1$ also cause gradient diffusion. On this condition, setting $w^{(2)} = 0$ can be a good solution for avoiding large output and restricting gradients in a suitable norm. Besides, it is feasible to update $w^{(2)}$ with the first non-trial step

$$w^{(2)} = -\lambda L w^{(1)} x^{(0)} \partial_x R(x)|_{x=x^{(L)}}, \qquad (18)$$

and will converge with a learning rate that is polynomial in the depth $L$ of the network. We plot the training dynamics in Figure 17, and use this simple example to illustrate the mechanism of IDInit, which is always a well-conditioned position for training.

# F TEMPORARY SECTION FOR REBUTTAL

## F.1 EXPERIMENT FOR HYPERPARAMETERS

In this experiment, we compare IDInit with other initialization methods, including (1) Fixup; (2) ReZero; (3) Kaiming; and (4) Zero, by analyzing the training hyperparameters, i.e., the weight decay and the learning rate. We use Cifar10. The backbone is ResNet-32, we use SGD with a momentum of 0.9. The batch size is 1024. We train models for 200 epochs. The learning rate is reduced with a cosine function. Each setting is trained 3 times to calculate the standard deviation.

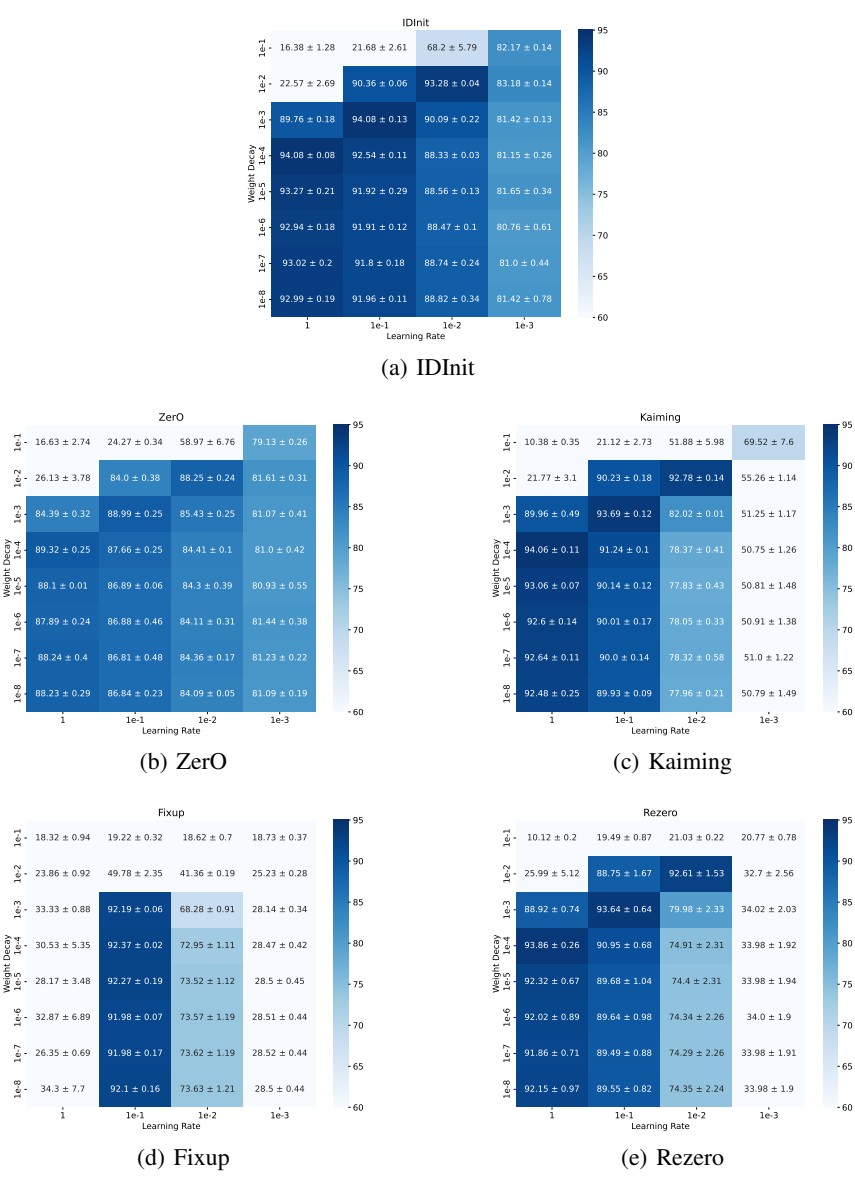

Figure 18: The hyperparameter experiment on Cifar10.

As shown in Figure 18, IDInit achieves a peak accuracy of 94.08% with a weight decay of 1e-3 and a learning rate of 1e-1. In comparison to other initialization methods including Kaiming, Fixup, and Rezero, IDInit demonstrates superior stability, maintaining high accuracy even when the learning rate is reduced below 1e-1. Although ZerO exhibits comparable stability at lower learning rates owing to its Hadamard matrix's ability to sustain dynamics, it underperforms at higher learning rates due to the dead neurons caused by the zero weights in its residual stems. Fixup, on the other

hand, lacks stability by eliminating batch normalization, rendering it unsuitable for high learning rates. Overall, IDInit consistently delivers robust performance while maintaining stability, making it a promising candidate for practical applications.

## F.2 ANALYSIS ON CONVERGENCE

The issue of convergence was proposed by Bartlett et al. (2019). According to their study, when layers in a neural network are initialized using the identity matrix, all the weight matrix of layers will be symmetric at each step of the training process. This persistent symmetry leads to the weights of layers being the same as each other at any step, posing a significant challenge in converging to the ground truth of which eigenvalues with negative values. Our findings indicate that employing a stochastic gradient descent (SGD) approach can effectively break the symmetry which facilitates convergence, and incorporating momentum can further accelerate the convergence process. In this context, we provide a formal proof demonstrating that SGD with momentum contributes to alleviating the convergence issue.

*Proof.* First of all, we present a training case for a single-layer network expressed as $Y = XW$, where $X \in \mathbb{R}^d$ represents the input, $Y \in \mathbb{R}^d$ denotes the output, and $W \in \mathbb{R}^{d \times d}$ is the weight matrix. The weight matrix $W$ is initialized to the identity matrix $I$, denoted as $W^{(0)} = I$. For our loss function, we employ the Mean Squared Error (MSE) and a learning rate denoted by $\eta$. Consider two training pairs $\{X_1, Y_1\}$ and $\{X_2, Y_2\}$ sampled from the same dataset $\mathcal{D}$. The network is initially trained with $\{X_1, Y_1\}$, and trained with $\{X_2, Y_2\}$ in the next step.

In the first step, we can get the prediction as

$$\hat{Y}_1 = X_1 W^{(0)}. \tag{19}$$

The updated $W^{(1)}$ can be derived by

$$\Delta W^{(0)} = X_1^T(\hat{Y}_1 - Y_1) = X_1^T(X_1 W^{(0)} - Y_1) = X_1^T(X_1 - Y_1),$$
$$W^{(1)} = W^{(0)} - \eta \Delta W^{(0)} = W^{(0)} - \eta X_1^T(X_1 - Y_1) = I - \eta X_1^T(X_1 - Y_1). \tag{20}$$

Therefore, in the second step, the gradient $\Delta W^{(1)}$ can be calculated as

$$\begin{aligned}
\Delta W^{(1)} &= X_2^T(\hat{Y}_2 - Y_2), \\
&= X_2^T(X_2 W^{(1)} - Y_2), \\
&= X_2^T(X_2(I - \eta X_1^T(X_1 - Y_1))) - Y_2), \\
&= X_2^T X_2 - \eta X_2^T X_2 X_1^T X_1 + \eta X_2^T X_2 X_1^T Y_1 - X_2^T Y_2. 
\end{aligned} \tag{21}$$

While $X_2^T X_2$ is symmetric, the component $-\eta X_2^T X_2 X_1^T X_1 + \eta X_2^T X_2 X_1^T Y_1 - X_2^T Y_2$ will be asymmetric. As $\eta$ is usually $1e-1$, and both training pairs $\{X_1, Y_1\}$ and $\{X_2, Y_2\}$ can be generally normalized to $\mathcal{N} \sim (0, 1)$, thereby, such magnitude of the asymmetric component can sufficiently influence the symmetry of the weight as

$$W^{(2)} = W^{(1)} - \eta \Delta W^{(1)}. \tag{22}$$

When introducing a momentum $M^{(0)}$ initialized to $\Delta W^{(0)}$, assuming the coefficient of $M$ is $\gamma$, $W^{(2)}$ will be updated as

$$\begin{aligned}
M^{(1)} &= \gamma M^{(0)} + \eta \Delta W^{(1)}, \\
W^{(2)} &= W^{(1)} - M^{(1)} = W^{(1)} - \gamma M^{(0)} - \eta \Delta W^{(1)}. 
\end{aligned} \tag{23}$$

Therefore, momentum can promote the weight to become asymmetric by accumulating the asymmetry of gradients in steps and impact more when samples are increased.

As for networks of multiple layers, when their layers are asymmetric, each layer can be updated differently which breaks the convergence problem caused by the same gradients in each step (which is stated in Lemma 5 of Bartlett et al. (2019)). □

This proof primarily demonstrates that SGD with momentum can effectively resolve the issue of layers being the same in networks initialized with the identity matrix during training, which facilitates the convergence process. As illustrated in Figure 19, it is evident that layers trained using SGD are different from each other, with the momentum component amplifying the degree of this difference. By theoretically and empirically demonstrating that SGD with momentum can efficiently address this convergence problem, we hope this finding can offer valuable insights for the research community, encouraging further investigation into identity initialization and its significant role in model training.

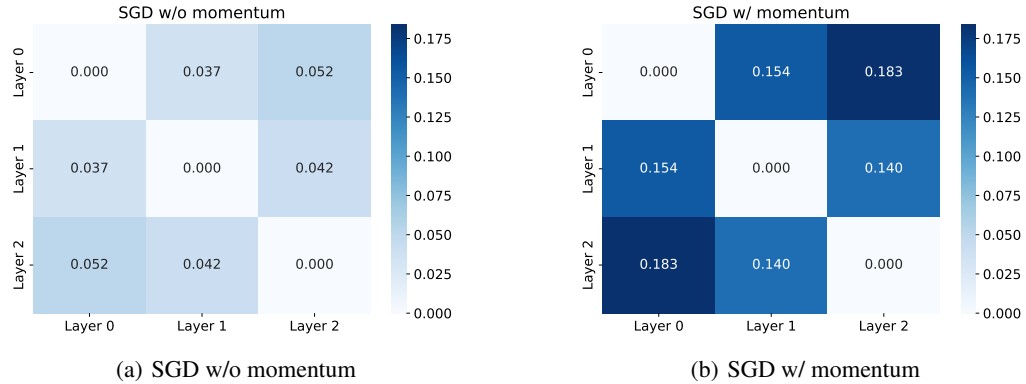

(a) SGD w/o momentum  (b) SGD w/ momentum

Figure 19: The distance between two layers in the network of Table 1 after training. We calculate the distance by averaging the absolute value from the difference value of two layers. Layers trained using SGD display distinct differences from one another, and the incorporation of momentum significantly increases these differences, thereby accelerating the convergence speed of the model.

