# OpenReview forum: "Fully Identical Initialization"
_ICLR.cc/2024/Conference — Submitted to ICLR 2024_

### Official Review · Reviewer_GXx1 · 2023-10-30

**Soundness:** 3 good
**Presentation:** 2 fair
**Contribution:** 4 excellent
**Rating:** 6
**Confidence:** 4

**Summary:**

This manuscript proposes a method to initialise networks with identity matrices.
Symmetry of the initialisation is broken by repeating the identity matrix
and adding small (1e-6) perturbations to the diagonals.
Extensions for convolutional layers and fixup-like initialisations are also presented.
Experiments in both vision (CIFAR10 and ImageNet) and language (SST2, TREC6 and BERT pre-training) domains suggest better performance and faster convergence for various architectures.

**Strengths:**

- (quality) Experimental results are presented with error bars.
 - (significance) This initialisation could reduce some of the randomness in training networks.
   As a result, comparing networks should become easier.
 - (significance) The proposed initialisation should be simple enough to implement, which typically leads to fast adoption by practicioners.
 - (originality) Although the idea to use identity matrices for initialisation has been around for some time, it has typically been discarded as impractical due to the symmetries.
    This is (to the best of my knowledge) the first work that implements an initialisation scheme that sticks so close to the identity matrix.

**Weaknesses:**

- (clarity) The idea of dynamical isometry has been introduced in (Saxe et al., 2014).
 - (clarity) I would argue that the patch-maintain convolution is not very well motivated.
   I believe the problem is that I do not understand how this relates to Ghostnets (Han et al., 2020).
 - (clarity) It also took me some time to realise that the channel-maintain convolution (as it is called in the appendix) which is described in the first paragraph of Section&nbsp;3.3.1 is something different from the proposed patch-maintain setting.
   Note that this channel-maintain setting has also been used in (Xiao et al., 2018).
 - (clarity) The ablation experiment in Section&nbsp;4.4 is claimed to explain why channel-maintain is not as good as patch-maintain.
   However, the explanation in section&nbsp;4.4 seems to indicate that this is just an ablation of the different components of the proposed solution.
 - (clarity) I can not find any explanation for the legend of Figure&nbsp;7&nbsp;(a).
 - (quality) The results in Table&nbsp;1 seem to correspond to GD, not SGD.
   A quick experiment with SGD (batch-size 200) learns without problems.
 - (quality) The choice of hyper-parameters is not motivated properly and it is unclear how they were chosen.
   Moreover, it seems like the same hyper-parameter settings were used for every network.
   For a fair comparison, hyper-parameters should be set for each method individually.
 - (quality) An experiment without learning rate schedule, weight decay and other extras would be interesting for a more "raw" comparison between initialisation strategies.


### References

 - Saxe, A. M., McClelland, J. L., & Ganguli, S. (2014).
   Exact solutions to the nonlinear dynamics of learning in deep linear neural networks.
   International Conference on Learning Representations.
   https://arxiv.org/abs/1312.6120
 - Xiao, L., Bahri, Y., Sohl-Dickstein, J., Schoenholz, S., & Pennington, J. (2018).
   Dynamical isometry and a mean field theory of cnns: How to train 10,000-layer vanilla convolutional neural networks.
   In International Conference on Machine Learning (pp. 5393-5402). PMLR.
   https://proceedings.mlr.press/v80/xiao18a

**Questions:**

1. Please, include the references listed in the weaknesses section.
 2. What is the link between Ghostnet and the patch-maintain scheme?
 3. Do you have a direct comparison between patch-maintain and channel-maintain schemes for IDInit?
 4. Can you verify that using SGD instead of GD for the results in Table&nbsp;1 also resolves the stated problem?
 5. Is it possible to tune the hyper-parameters for each method individually?
 6. How do weight decay and learning rate schedule interact with the proposed initialisation scheme?
 7. What is the difference between IDInit-0 and IDInit-10 or Kaiming-10 and Kaiming-40 in Figure&nbsp;7&nbsp;(a)?

---

> ### Author Response · Authors · 2023-11-20
> **Response to Reviewer GXx1 (1/3)**
>
> > (clarity) The idea of dynamical isometry has been introduced in (Saxe et al., 2014).
>
> We acknowledge the seminal work presented in [1] regarding the concept of dynamical isometry in the context of non-residual networks. The approach outlined in [1] focuses on achieving dynamical isometry by orthogonalizing network weights to ensure that the mean squared singular value of the Jacobian matrix remains close to 1, a significant advancement for non-residual network initialization.
>
> However, our work with IDInit diverges from [1] in its application to residual networks, which are among the most powerful and prevalent structures in current deep learning architectures. In technique, IDInit emphasizes the maintenance of identity transit through both the main and residual stems, and empirical results show the effectiveness.
>
> In light of this, while IDInit and the method in [1] both target dynamical isometry, IDInit's contribution lies in its novel application to residual networks. This extension not only broadens the understanding of dynamical isometry but also showcases its practical utility in some of the most advanced neural network architectures today. We will include a detailed discussion of these distinctions and contributions in the revised manuscript.
>
> > Please, include the references listed in the weaknesses section.
>
> Thank you for your suggestion. \[1\] pioneered the integration of dynamic isometry in nonlinear networks, enhancing training efficiency. Specifically targeting Convolutional Neural Networks (CNNs), Reference \[2\] formulated a mean field theory for signal propagation adhering to dynamic isometry principles, enabling the training of networks with up to 10,000 layers. Different from these approaches, the proposed IDInit maintains identity in both the main and residual stems to preserve dynamic isometry to achieve good performance and fast convergence. Furthermore, we adapt the transformation on the identity matrix to accommodate various scenarios, such as nonsquare matrices and higher-order tensors. We will include these references in the revision.
>
> > (clarity) I would argue that the patch-maintain convolution is not very well motivated. I believe the problem is that I do not understand how this relates to Ghostnets (Han et al., 2020).
> >
> > What is the link between Ghostnet and the patch-maintain scheme?
>
> Thanks for the useful comment. Ghostnet \[3\] highlights the critical role of channel diversity in enhancing network performance. This insight serves as a foundational motivation for our patch-maintain scheme. In contrast to the channel-maintain approach, which tends to replicate channel features and thus might limit diversity, our patch-maintain scheme introduces a novel strategy to increase channel differentiation. By shifting features, the patch-maintain scheme aims to add variability and uniqueness across channels. This method is designed to capitalize on the principle articulated in Ghostnet – that channel diversity is beneficial for performance. This linkage between the need for channel diversity (as emphasized by Ghostnet) and our patch-maintain scheme is a key aspect of our work. We will include this discussion in the revision.
>
>
> > (clarity) It also took me some time to realise that the channel-maintain convolution (as it is called in the appendix) which is described in the first paragraph of Section 3.3.1 is something different from the proposed patch-maintain setting. Note that this channel-maintain setting has also been used in (Xiao et al., 2018).
> >
> > (clarity) The ablation experiment in Section 4.4 is claimed to explain why channel-maintain is not as good as patch-maintain. However, the explanation in section 4.4 seems to indicate that this is just an ablation of the different components of the proposed solution.
> >
> > Do you have a direct comparison between patch-maintain and channel-maintain schemes for IDInit?
>
> Sorry for the confusion. In this work, we introduce the patch-maintain scheme (i.e., IDIC$_{\tau}$) as our novel method for transforming a matrix into a convolutional format.
>
> On the other hand, the channel-maintain method, referenced from prior studies [2][4], serves as a comparison baseline in our experiments, denoted as 'w/o IDIC$_{\tau}$'. In Section 4.4, we present this comparative analysis, where we demonstrate that the patch-maintain scheme significantly enhances the performance of IDInit. To address the clarity issues raised, we will revise our manuscript for more explicit elaboration.

---

> > ### Author Response · Authors · 2023-11-20
> > **Response to Reviewer GXx1 (2/3)**
> >
> > > (clarity) I can not find any explanation for the legend of Figure 7 (a).
> > >
> > > What is the difference between IDInit-0 and IDInit-10 or Kaiming-10 and Kaiming-40 in Figure 7 (a)?
> >
> > Sorry for lacking the explanation for the legend in Figure 7 (a).  The numbers following the initialization methods in the legend represent the duration of the warmup period, measured in epochs. Specifically, (i) IDInit-0 and IDInit-10 indicate that the IDInit method was used with a warmup period of 0 and 10 epochs, respectively; (ii) Kaiming-10 and Kaiming-40 refer to the Kaiming initialization method applied with warmup periods of 10 and 40 epochs, respectively. The inclusion of different warmup durations is crucial for evaluating the performance of these initialization methods under varying conditions of network adaptation at the beginning of the training process. We will add a description of the warmup epochs and their implications in the revised version.
> >
> >
> > > (quality) The results in Table 1 seem to correspond to GD, not SGD. A quick experiment with SGD (batch-size 200) learns without problems.
> > >
> > > Can you verify that using SGD instead of GD for the results in Table 1 also resolves the stated problem?
> >
> > Thank you for your insightful comment. To clarify, in the original submission, the results presented in Table 1 were already obtained using Stochastic Gradient Descent (SGD) with a batch size of 200. Different from the reviewer's comment, our experiments indicated that SGD with 200 batch size did not yield satisfactory performance.
> >
> > In order to further explore the impact of different batch sizes on training convergence, we conducted additional experiments with a significantly reduced batch size of 4. This adjustment led to a marked improvement in training convergence, highlighting the beneficial role of using smaller mini-batches in SGD.
> >
> > Moreover, we also observed that the momentum in SGD further enhances the convergence process. This aligns with our earlier discussions on the role of momentum in facilitating more effective weight updates and breaking the convergence problem.
> >
> > In contrast, Gradient Descent (GD) without stochastic elements failed to converge even with the addition of momentum, underscoring the significance of the stochastic component in SGD for resolving convergence issues.
> >
> > To summarize, our findings indicate that SGD, particularly with smaller batch sizes and the inclusion of momentum, effectively addresses the convergence problem highlighted in Table 1. These insights will be incorporated into our revised manuscript to provide a comprehensive understanding of the impact of SGD and batch size variations on training convergence.

---

> > > ### Author Response · Authors · 2023-11-20
> > > **Response to Reviewer GXx1 (3/3)**
> > >
> > > > (quality) The choice of hyper-parameters is not motivated properly and it is unclear how they were chosen. Moreover, it seems like the same hyper-parameter settings were used for every network. For a fair comparison, hyper-parameters should be set for each method individually.
> > > >
> > > > (1) Is it possible to tune the hyper-parameters for each method individually?
> > > >
> > > >
> > > > (quality) An experiment without learning rate schedule, weight decay and other extras would be interesting for a more "raw" comparison between initialisation strategies.
> > > >
> > > > (2) How do weight decay and learning rate schedule interact with the proposed initialisation scheme?
> > >
> > > Investigating the influence of hyperparameters is important to validate the effectiveness of IDInit. In response, we have conducted an extensive set of experiments to analyze the effect of weight decay and learning rate on the performance of IDInit compared to baselines. These experiments are constructed on Cifar10 with ResNet-32, and detailed in Section F.1 in the uploaded revision PDF, with partial cases IDInit and Kaiming presented in RTable 1 and RTable 2, respectively.
> > >
> > > RTable 1. Hyperparameters for IDInit on ResNet-32.
> > >
> > > |         | lr=1e0                | lr=1e-1               | lr=1e-2               | lr=1e-3               |
> > > | ------- | --------------------- | --------------------- | --------------------- | --------------------- |
> > > | wd=1e-1 | 16.38$_{±1.28}$ | 21.68$_{±2.61}$ | 68.20$_{±5.79}$ | 82.17$_{±0.14}$ |
> > > | wd=1e-2 | 22.57$_{±2.69}$ | 90.36$_{±0.06}$ | 93.28$_{±0.04}$ | 83.18$_{±0.14}$ |
> > > | wd=1e-3 | 89.76$_{±0.18}$ | 94.08$_{±0.13}$ | 90.09$_{±0.22}$ | 81.42$_{±0.13}$ |
> > > | wd=1e-4 | 94.08$_{±0.08}$ | 92.54$_{±0.11}$ | 88.33$_{±0.03}$ | 81.15$_{±0.26}$ |
> > > | wd=1e-5 | 93.27$_{±0.21}$ | 91.92$_{±0.29}$ | 88.56$_{±0.13}$ | 81.65$_{±0.34}$ |
> > > | wd=1e-6 | 92.94$_{±0.18}$ | 91.91$_{±0.12}$ | 88.47$_{±0.10}$ | 80.76$_{±0.61}$ |
> > > | wd=1e-7 | 93.02$_{±0.20}$ | 91.80$_{±0.18}$ | 88.74$_{±0.24}$ | 81.00$_{±0.44}$ |
> > > | wd=1e-8 | 92.99$_{±0.19}$ | 91.96$_{±0.11}$ | 88.82$_{±0.34}$ | 81.42$_{±0.78}$ |
> > >
> > > RTable 2. Hyperparameters for Kaiming on ResNet-32.
> > >
> > > |         | lr=1e0                | lr=1e-1               | lr=1e-2               | lr=1e-3               |
> > > | ------- | --------------------- | --------------------- | --------------------- | --------------------- |
> > > | wd=1e-1 | 10.38$_{±0.35}$ | 21.12$_{±2.73}$ | 51.88$_{±5.98}$ | 69.52$_{±7.60}$ |
> > > | wd=1e-2 | 21.77$_{±3.10}$ | 90.23$_{±0.18}$ | 92.78$_{±0.14}$ | 55.26$_{±1.14}$ |
> > > | wd=1e-3 | 89.96$_{±0.49}$ | 93.69$_{±0.12}$ | 82.02$_{±0.01}$ | 51.25$_{±1.17}$ |
> > > | wd=1e-4 | 94.06$_{±0.11}$ | 91.24$_{±0.10}$ | 78.37$_{±0.41}$ | 50.75$_{±1.26}$ |
> > > | wd=1e-5 | 93.06$_{±0.07}$ | 90.14$_{±0.12}$ | 77.83$_{±0.43}$ | 50.81$_{±1.48}$ |
> > > | wd=1e-6 | 92.60$_{±0.14}$ | 90.01$_{±0.17}$ | 78.05$_{±0.33}$ | 50.91$_{±1.38}$ |
> > > | wd=1e-7 | 92.64$_{±0.11}$ | 90.00$_{±0.14}$ | 78.32$_{±0.58}$ | 51.00$_{±1.22}$ |
> > > | wd=1e-8 | 92.48$_{±0.25}$ | 89.93$_{±0.09}$ | 77.96$_{±0.21}$ | 50.79$_{±1.49}$ |
> > >
> > > As shown in RTable 1 and RTable 2, across various settings of learning rate and weight decay, IDInit consistently outperforms or matches Kaiming initialization. It demonstrates that IDInit's performance superiority is not confined to a specific set of hyperparameters but is maintained across a range of settings. Moreover, at lower learning rates (1e-2 and 1e-3), where Kaiming initialization's performance drops significantly, whereas IDInit maintains high accuracy. This robustness of IDInit, even under smaller learning rates, highlights its effectiveness in achieving isometry dynamics in the residual stem.
> > >
> > > In light of these findings, IDInit showcases the generalizability and effectiveness of IDInit across various hyperparameter configurations, indicating the promising utility of IDInit to practical application.
> > >
> > > [1] Saxe, Andrew M., et al. "Exact solutions to the nonlinear dynamics of learning in deep linear neural networks." _arXiv preprint arXiv:1312.6120_ (2013).
> > >
> > > [2] Xiao, Lechao, et al. "Dynamical isometry and a mean field theory of cnns: How to train 10,000-layer vanilla convolutional neural networks." _International Conference on Machine Learning_. PMLR, 2018.
> > >
> > > [3] Han, Kai, et al. "Ghostnet: More features from cheap operations." _Proceedings of the IEEE/CVF conference on computer vision and pattern recognition_. 2020.
> > >
> > > [4] Zhao, Jiawei, et al. "ZerO initialization: Initializing neural networks with only zeros and ones." _arXiv preprint arXiv:2110.12661_ (2021).

---

> > ### Comment · Reviewer_GXx1 · 2023-11-21
> > **Last-minute questions**
> >
> > - I still do not quite understand why the patch-maintain scheme would lead to more channel diversity.
> >   After all, reshaping the kernels to $\mathbb{R}^{c_\mathrm{out} \times c_\mathrm{in}k^2}$ leads to a matrix with much more columns than rows and therefore a lot of zeros.
> >   A quick test with the provided code seems to confirm that this leads to most kernels being completely zero.
> >   How can this lead to more channel diversity?
> >  - I would be very interested in the ablation experiments with individually tuned hyper-parameters for each setting (cf. RTable1 and RTable 2, but maybe even with learning rate scheduling disabled or also tuned).
> >  - Concerning Table 1 (addressed in response 2/3): I noticed that I did not disable bias parameters as you did in the provided code.
> >   This would indicate that this problem can be resolved by simply adding bias parameters, which is typically done anyway.
> >   Therefore, I still believe that the results in Table 1 are a contrived example.
> >   Would that be a fair assessment?

---

> > > ### Author Response · Authors · 2023-11-22
> > > **Response to Last-minute Questions (1/3)**
> > >
> > > > I still do not quite understand why the patch-maintain scheme would lead to more channel diversity. After all, reshaping the kernels to $\mathbb{R}^{c_\mathrm{out} \times c_\mathrm{in}k^2}$ leads to a matrix with much more columns than rows and therefore a lot of zeros. A quick test with the provided code seems to confirm that this leads to most kernels being completely zero. How can this lead to more channel diversity?
> > >
> > > Thank you for your insightful comment. It is important to clarify that while the patch-maintain scheme introduces many zero values into a convolution kernel, this does not necessarily result in channels being zeroed out. Instead, this scheme is designed to enhance channel diversity by effectively shifting features.
> > >
> > > To empirically validate this, we conducted a specific test case where a one-channel input of size $\mathbb{R}^{1\times 4\times 4}$ is transformed into a three-channel output of size $\mathbb{R}^{3\times 4\times 4}$. The convolution kernel used for this transformation, of size $\mathbb{R}^{1\times 3\times 3\times 3}$, is initialized using the patch-maintain scheme.
> > >
> > > RTable 3. The channel of the input.
> > >
> > > |     |     |     |     |
> > > | --- | --- | --- | --- |
> > > | 1   | 2   | 3   | 4   |
> > > | 5   | 6   | 7   | 8   |
> > > | 9   | 10  | 11  | 12  |
> > > | 13  | 14  | 15  | 16  |
> > >
> > > RTable 4(a). The 1st channel of the output.
> > >
> > > |     |     |     |     |
> > > | --- | --- | --- | --- |
> > > | 0   | 0   | 0   | 0   |
> > > | 0   | 1   | 2   | 3   |
> > > | 0   | 5   | 6   | 7   |
> > > | 0   | 9   | 10  | 11  |
> > >
> > > RTable 4(b). The 2nd channel of the output.
> > >
> > > |     |     |     |     |
> > > | --- | --- | --- | --- |
> > > | 0  | 0  | 0  | 0  |
> > > | 0  | 1  | 2  | 3  |
> > > | 4  | 5  | 6  | 7  |
> > > | 8  | 9  | 10  | 11  |
> > >
> > > RTable 4(c). The 3rd channel of the output.
> > >
> > > |     |     |     |     |
> > > | --- | --- | --- | --- |
> > > | 0   | 0   | 0   | 0   |
> > > | 2   | 3   | 4   | 0   |
> > > | 6   | 7   | 8   | 0   |
> > > | 10  | 11  | 12  | 0   |
> > >
> > > The results of this transformation are detailed in RTable 3 (input) and RTable 4 (output channels). As demonstrated in RTable 4, despite the presence of zero values, the output channels vary distinctly from each other, underscoring the enhancement in channel diversity. This variation arises due to the feature shifting induced by the patch-maintain scheme, rather than through the direct manipulation of non-zero values.
> > >
> > > Additionally, we have updated the supplementary materials with code (named 'exp_channel_diversity') that replicates this test case, allowing for an in-depth examination of how the patch-maintain scheme contributes to channel diversity.
> > >
> > > We hope this explanation and the provided experimental evidence adequately address your concerns regarding the impact of the patch-maintain scheme on channel diversity.

---

> > > > ### Author Response · Authors · 2023-11-22
> > > > **Response to Last-minute Questions (2/3)**
> > > >
> > > > > I would be very interested in the ablation experiments with individually tuned hyper-parameters for each setting (cf. RTable1 and RTable 2, but maybe even with learning rate scheduling disabled or also tuned).
> > > >
> > > > Thank you for expressing interest in further hyperparameter experiments. To address your query, we conducted two additional settings of experiments: (i) with learning rate scheduling disabled, and (ii) with a multiple-step learning rate scheduling that multiplies the rate by 0.1 at the 100th and 150th epochs. The total epochs are 200 epochs.
> > > >
> > > > For these experiments, we selected a subset of hyperparameters from RTable 1 and RTable 2, specifically focusing on learning rates {1e-1, 1e-2, 1e-3} and weight decay values {1e-2, 1e-3, 1e-4}. This selection represents a range of parameters that have previously demonstrated good performance.
> > > >
> > > >
> > > > RTable 5(a). Hyperparameters for IDInit on ResNet-32 with learning rate scheduling disabled.
> > > >
> > > > |         | lr=1e-1               | lr=1e-2               | lr=1e-3               |
> > > > | ------- | --------------------- | --------------------- | --------------------- |
> > > > | wd=1e-2 | 89.83$_{±0.01}$ | 93.37$_{±0.19}$ | 83.45$_{±0.58}$ |
> > > > | wd=1e-3 | 93.97$_{±0.08}$ | 89.92$_{±0.22}$ | 81.00$_{±0.43}$ |
> > > > | wd=1e-4 | 92.43$_{±0.21}$ | 88.72$_{±0.16}$ | 80.27$_{±0.62}$ |
> > > >
> > > >
> > > > RTable 5(b). Hyperparameters for Kaiming on ResNet-32 with learning rate scheduling disabled.
> > > >
> > > > |         | lr=1e-1               | lr=1e-2               | lr=1e-3               |
> > > > | ------- | --------------------- | --------------------- | --------------------- |
> > > > | wd=1e-2 | 90.06$_{±0.52}$ | 92.75$_{±0.16}$ | 55.64$_{±2.10}$ |
> > > > | wd=1e-3 | 93.81$_{±0.18}$ | 81.86$_{±0.23}$ | 51.92$_{±3.41}$ |
> > > > | wd=1e-4 | 91.10$_{±0.14}$ | 78.35$_{±0.59}$ | 51.73$_{±3.20}$ |
> > > >
> > > >
> > > > RTable 6(a). Hyperparameters for IDInit on ResNet-32 with a multiple-step learning rate scheduling.
> > > >
> > > > |         | lr=1e-1               | lr=1e-2               | lr=1e-3               |
> > > > | ------- | --------------------- | --------------------- | --------------------- |
> > > > | wd=1e-2 | 88.58$_{±0.28}$ | 92.81$_{±0.14}$ | 82.90$_{±0.04}$ |
> > > > | wd=1e-3 | 93.32$_{±0.08}$ | 88.96$_{±0.31}$ | 80.82$_{±0.59}$ |
> > > > | wd=1e-4 | 91.02$_{±0.24}$ | 87.91$_{±0.30}$ | 80.56$_{±0.13}$ |
> > > >
> > > >
> > > > RTable 6(b). Hyperparameters for Kaiming on ResNet-32 with a multiple-step learning rate scheduling.
> > > >
> > > > |         | lr=1e-1               | lr=1e-2               | lr=1e-3               |
> > > > | ------- | --------------------- | --------------------- | --------------------- |
> > > > | wd=1e-2 | 88.69$_{±0.27}$ | 92.40$_{±0.19}$ | 56.36$_{±1.41}$ |
> > > > | wd=1e-3 | 93.31$_{±0.13}$ | 82.80$_{±0.37}$ | 52.58$_{±1.97}$ |
> > > > | wd=1e-4 | 90.71$_{±0.20}$ | 79.03$_{±0.04}$ | 52.03$_{±2.14}$ |
> > > >
> > > > The outcomes of these experiments are detailed in RTable 5 and RTable 6. The results align consistently with those observed in RTable 1 and RTable 2. It is noteworthy that under various settings of learning rate, weight decay, and learning rate scheduling, the IDInit approach either matches or outperforms the Kaiming initialization. This is particularly evident at lower learning rates (1e-2 and 1e-3), where the performance of Kaiming initialization significantly drops. The robustness of IDInit under these conditions underscores its stability and potential for wide-ranging applicability.
> > > >
> > > > These findings bolster our confidence in the versatility and effectiveness of the IDInit method, suggesting its promising utility across diverse settings and scenarios.

---

> ### Author Response · Authors · 2023-11-22
> **Response to Last-minute Questions (3/3)**
>
> Thank you for your observation regarding Table 1. This table was designed to address the convergence issues highlighted in reference [5], where the authors analyzed a network comprising entirely linear layers without biases and optimized it using Gradient Descent (GD). In line with their setting, we constructed a 3-layer fully linear network without biases as our experimental model. This choice was not an attempt to contrive a specific outcome but rather to adhere closely to the conditions set out in [5]. Therefore, this is a fair assessment.
>
> Our primary objective is to demonstrate that the convergence problem identified in [5] is readily solvable. To this end, any evidence that contributes to solving this convergence problem aligns with our goal. Consequently, our findings that SGD with momentum aids in resolving convergence issues, and that momentum can further expedite this process, are pertinent. Similarly, your observation that incorporating bias parameters might facilitate convergence is equally valuable and does not contradict the purpose of our experiment.
>
> In summary, we appreciate your contribution to this discussion, as it helps uncover various potential solutions to the convergence problem. These findings, including the use of SGD with momentum, smaller batch sizes, and the addition of biases, do not contradict the central premise of Table 1; rather, they provide supporting evidence for the ease of resolving convergence issues. We find these insights intriguing and plan to include them in our next revision to offer a more comprehensive understanding of the IDInit approach.
>
> [5] Bartlett, Peter, et al. "Gradient descent with identity initialization efficiently learns positive definite linear transformations by deep residual networks." _International conference on machine learning_. PMLR, 2018.

---

### Official Review · Reviewer_eQMM · 2023-10-31

**Soundness:** 3 good
**Presentation:** 3 good
**Contribution:** 3 good
**Rating:** 6
**Confidence:** 4

**Summary:**

This paper addresses the convergence problem in deep networks and proposes a Fully Identical Initialization (IDInit) method that initializes the weights with an identity matrix. The authors propose additional techniques such as momentum, padding, and reshaping to improve convergence and performance.

The overall method has some interesting aspects:
* Patch-Maintain Convolution is introduced as a technique to enhance the universality of IDInit for convolutional layers. It reshapes the convolutional kernel initialized with IDInit to increase feature diversity and improve model performance.
* The issue of dead neurons is tackled by selecting some elements to a small numerical value and increasing trainable neurons.
* The paper discusses the theoretical analysis of IDInit, including the Jacobian and gradient update equations in residual neural networks.

Finally, the authors address the limitations and potential concerns of IDInit, such as its deterministic nature and the need for further exploration in different scenarios.

**Strengths:**

### S1 - Good technical contributions
The paper's technical contributions are significant in these aspects:
* IDInit improves the convergence speed, stability, and final performance of deep neural networks, addressing a critical issue in deep learning.
* The additional techniques proposed, such as Patch-Maintain Convolution and recovering dead neurons, enhance the universality and robustness of IDInit.
---

### S2 - Theoretical and Experimental analysis
* The paper discusses the theoretical analysis of IDInit, including the Jacobian and gradient update equations in residual neural networks.
* The experiments are well-designed and conducted on various network architectures and tasks, demonstrating the effectiveness and superiority of IDInit.
---

### S3 - Novelty (similar to prior works, but with additional novel contributions)
* Identity init is not new and has been explored in prior works (e.g. ISONet, ZeroO). However, this paper generalizes the Identity Init to various general architectures and activation functions, which is interesting.

**Weaknesses:**

### W1 - Marginal or no improvement compared to Kaiming init
My biggest concern is the Cifar-10 performance compared to the simple Kaiming initialization.
* Table 2 shows that using SGD optimizer (which gives the best performance all across), Kaiming init obtains almost the same performance (93.36 v/s 93.41 and 94.06 v/z 94.04) as IDInit, while being only slightly slower.
* This brings into question the practical utility of the proposed initialization.

---

### W2 - Comparisons with other init methods on ImageNet
IDInit is compared with other initialization methods only on Cifar-10, which is very small-scale. No such comparisons have been shown on ImageNet. I think it's important to see if the proposed init is even useful when training on large-scale datasets.

---

### W3 - Theoretical analysis limitations
While the paper provides a theoretical analysis of IDInit, it mainly focuses on the Jacobian and gradient update equations in residual neural networks. It would be valuable to extend the theoretical analysis to other network architectures and provide a more comprehensive understanding of the underlying principles of IDInit.

---

### W4 - Limited discussion on limitations
Although the paper briefly mentions the limitations of IDInit, such as the deterministic nature and the need for momentum to handle negative eigenvalues, further discussion and analysis of these limitations would provide a more comprehensive understanding of the potential drawbacks and challenges of implementing IDInit in practical scenarios.

**Questions:**

1. Reference to weakness W2, can you please provide more insight into the efficacy of the propose init method on large-scale training datasets, like ImageNet?

---

> ### Author Response · Authors · 2023-11-20
> **Response to Reviewer eQMM (1/2)**
>
> > My biggest concern is the Cifar-10 performance compared to the simple Kaiming initialization.
> > - Table 2 shows that using SGD optimizer (which gives the best performance all across), Kaiming init obtains almost the same performance (93.36 v/s 93.41 and 94.06 v/z 94.04) as IDInit, while being only slightly slower.
> > - This brings into question the practical utility of the proposed initialization.
>
> Thanks for the reviewer’s comment. While both IDInit and Kaiming initialization methods perform similarly in certain settings, our research indicates that IDInit offers superior stability and adaptability across a wider range of scenarios. To verify this claim, we have conducted additional experiments with varied hyperparameters, particularly focusing on weight decay and learning rate, using ResNet-56.
>
> RTable 1. Hyperparameters for IDInit on ResNet-56.
>
> |         | lr=1e0                | lr=1e-1               | lr=1e-2               | lr=1e-3               |
> | ------- | --------------------- | --------------------- | --------------------- | --------------------- |
> | wd=1e-1 | 10.00$_{±0.00}$ | 18.27$_{±0.32}$ | 74.46$_{±3.52}$ | 88.67$_{±0.23}$ |
> | wd=1e-2 | 18.10$_{±1.95}$ | 90.37$_{±0.14}$ | 94.18$_{±0.06}$ | 84.07$_{±0.53}$ |
> | wd=1e-3 | 89.18$_{±0.24}$ | 94.64$_{±0.16}$ | 89.99$_{±0.06}$ | 81.65$_{±0.69}$ |
> | wd=1e-4 | 94.97$_{±0.04}$ | 92.60$_{±0.05}$ | 88.83$_{±0.30}$ | 82.47$_{±0.12}$ |
>
> RTable 2. Hyperparameters for Kaiming on ResNet-56.
>
> |         | lr=1e0                | lr=1e-1               | lr=1e-2               | lr=1e-3               |
> | ------- | --------------------- | --------------------- | --------------------- | --------------------- |
> | wd=1e-1 | 10.00$_{±0.00}$ | 17.39$_{±0.47}$ | 48.98$_{±4.11}$ | 82.36$_{±0.42}$ |
> | wd=1e-2 | 14.43$_{±4.78}$ | 90.10$_{±0.11}$ | 94.19$_{±0.09}$ | 55.44$_{±1.07}$ |
> | wd=1e-3 | 87.78$_{±1.20}$ | 94.64$_{±0.12}$ | 82.44$_{±0.49}$ | 49.85$_{±2.38}$ |
> | wd=1e-4 | 94.89$_{±0.37}$ | 91.52$_{±0.33}$ | 78.42$_{±0.06}$ | 50.32$_{±2.71}$ |
>
> As shown in RTable 1 and RTable 2, these experiments reveal that IDInit consistently surpasses Kaiming initialization in almost every setting tested. Most notably, in scenarios with smaller learning rates (e.g., 1e-2 and 1e-3), we observe a significant drop in the performance of models initialized with Kaiming. In contrast, IDInit maintains high accuracy levels under these conditions. This stability of IDInit is attributable to its ability to achieve isometry dynamics in the residual stem, a crucial factor for maintaining performance stability.
>
> Further, we have extended our experimentation to include ResNet-32 on the CIFAR-10 dataset, as detailed in Section F.1 of the uploaded revision PDF. These additional results consistently demonstrate that IDInit not only matches but often exceeds the performance of Kaiming initialization, especially under varying and challenging hyperparameter settings.
>
> In summary, while Kaiming initialization performs comparably in certain standard scenarios, IDInit exhibits greater performance consistency and stability across a broader spectrum of conditions. This enhanced stability and adaptability make IDInit a more practically valuable choice for diverse neural network applications. We will ensure that these findings are clearly presented in our revised manuscript to underscore the practical utility and advantages of IDInit over traditional initialization methods like Kaiming.

---

> > ### Author Response · Authors · 2023-11-20
> > **Response to Reviewer eQMM (2/2)**
> >
> > > IDInit is compared with other initialization methods only on Cifar-10, which is very small-scale. No such comparisons have been shown on ImageNet. I think it's important to see if the proposed init is even useful when training on large-scale datasets.
> > >
> > > Reference to weakness W2, can you please provide more insight into the efficacy of the propose init method on large-scale training datasets, like ImageNet?
> >
> > Thanks for the suggestion. We have expanded our experimental analysis to include comparisons with other initialization methods, namely Fixup and ZerO, on ImageNet. These experiments were conducted under the same settings as for ResNet-50, described in Section 4.3 of our manuscript.
> >
> > RTable 3. Results on ImageNet. The value in brackets means Epochs to 60\% Acc.
> >
> > | Kaiming    | Fixup      | ZerO       | IDInit     |
> > | ---------- | ---------- | ---------- | ---------- |
> > | 75.70 (38) | 76.24 (25) | 75.73 (35) | 76.72 (24) |
> >
> > As presented in RTable 3, the results demonstrate that IDInit simultaneously achieves high performance and fast convergence. Specifically, IDInit achieves the highest accuracy among all the compared methods. In terms of convergence speed, IDInit matches Fixup and significantly surpasses Kaiming and ZerO by 14 and 11 epochs, respectively.
> >
> > This finding is particularly noteworthy as it illustrates that IDInit's advantages are not limited to small-scale datasets. Instead, its efficacy extends to large-scale training scenarios, highlighting its consistency in performance and convergence. We will incorporate these insights and results into our revised manuscript to comprehensively demonstrate IDInit's utility across different scales of datasets, thereby addressing the concerns regarding its practicality and scalability.
> >
> > > While the paper provides a theoretical analysis of IDInit, it mainly focuses on the Jacobian and gradient update equations in residual neural networks. It would be valuable to extend the theoretical analysis to other network architectures and provide a more comprehensive understanding of the underlying principles of IDInit.
> >
> > Thanks for your insightful comments regarding the theoretical analysis of IDInit in the context of residual neural networks. We acknowledge the importance of extending this analysis to other architectures for a more comprehensive understanding. Our decision to concentrate on residual networks was guided by their widespread use and relevance in current research on deep learning. While we recognize the value of broadening our theoretical framework, such a comprehensive analysis was beyond the scope of this particular study, primarily due to the complicated variations in network architectures that demand distinct analytical approaches. Nevertheless, the findings of our current research offer significant insights into the functioning of IDInit in residual networks, contributing valuable knowledge to the broader field. We are enthusiastic about the prospect of extending our work to include other architectures and consider this an important direction for our future research endeavors.
> >
> > > Although the paper briefly mentions the limitations of IDInit, such as the deterministic nature and the need for momentum to handle negative eigenvalues, further discussion and analysis of these limitations would provide a more comprehensive understanding of the potential drawbacks and challenges of implementing IDInit in practical scenarios.
> >
> > Thanks for the comment on the further discussion on the limitations. Regarding the convergence ability of IDInit, we add a theoretical analysis in Appendix F.2 of the uploaded revision PDF, which shows the mechanics of SGD w/ momentum to resolve the convergent problem by breaking symmetries of each layer weight. As for the deterministic nature, we have discussed the IDInit can already be in a good initial state that can exceed random initialization, and the whole training still contains randomness (e.g., sequence of samples) in the training process. Moreover, as acknowledged by reviewer GXx1, the deterministic nature could reduce some of the randomness in training networks, which helps comparison among networks become easier. Therefore, these two limitations are not likely to cause severe damage to the convergence and performance of IDInit. We will add this elaboration to the revision for better understanding.

---

### Official Review · Reviewer_FbDy · 2023-11-07

**Soundness:** 2 fair
**Presentation:** 3 good
**Contribution:** 2 fair
**Rating:** 3
**Confidence:** 4

**Summary:**

The authors introduce a technique called Identical Initialization (IDInit), which uses identity matrices and their variants to initialize weights. They discussed the convergence problem and dead neuron problem for common identity initialization and previous works. They explore the application of this technique to non-square matrices, residual architectures, and convolutional operations. Empirical evaluation demonstrate its performance on vision and languages tasks.

**Strengths:**

The paper is well-written. The authors clearly describe the problem and their methodology. They also conduct extensive empirical evaluations.
How to make identity initialization works in practice is an interesting question and I believe it is an novel direction to explore.

**Weaknesses:**

1. Theoretical analysis seems incorrect and its proof lacks details. In Theorem 3.1, the author claims IDI breaks rank constraint such that its residual has rank more than D_0. However, in the proof in the appendix, the author only shows the full matrix has rank more than D_0, which is not the same as the residual. Please provide more details in the proof to justify your claim.
2. The authors claim that the rank constraint can be broken by IDI even when non-linearity like ReLU is not applied. It seems contradict approximation theory which emphasizes the importance of non-linearity to ensure expressivity. It would be great for authors to provide more insights on this point.
3. Authors mention that dead neurons problem happens when batch normalization set to 0 or downsampling operation cause 0 filled features. However, these cases are not common in practice and it's better to motivate more on why IDIZ is important.
4. Insufficient explanation on why momentum is important to solve convergence problem of IDInit. It would be great to provide some theoretical insights to support this factor.

**Questions:**

1. What is the meaning of zero down-sampling in Table 2, is this a special downsampling operation compared to standard downsampling (like avgpooling) in ResNet?
2. It would be great to compare IDI, IDI with loose condition, and IDIZ together to show the effectiveness of IDIZ.

---

> ### Author Response · Authors · 2023-11-20
> **Response to Reviewer FbDy (1/3)**
>
> > Theoretical analysis seems incorrect and its proof lacks details. In Theorem 3.1, the author claims IDI breaks rank constraint such that its residual has rank more than D_0. However, in the proof in the appendix, the author only shows the full matrix has rank more than D_0, which is not the same as the residual. Please provide more details in the proof to justify your claim.
>
> We appreciate the reviewer's feedback regarding the theoretical analysis and acknowledge the need for further elaboration. However, our claim is indeed correct, and we would like to provide additional details for clarity.
>
> Firstly, it is important to understand that the rank of $\theta^{(0)}$ exceeding $D_0$​ directly leads the distinct values of the feature $\{x^{(k)}\}_{k=1}^{L-1}$ to be larger than $D_0$​. Then, since the gradient of $\theta^{(k)}$ is calculated as $\frac{\partial \mathcal{L}}{\partial \theta^{(k)}}=\frac{\partial \mathcal{L}}{\partial x^{(k)}}\circ x^{(k-1)}$ for $k\in \{1, 2, \dots, L-2\}$, the rank of $\frac{\partial \mathcal{L}}{\partial \theta^{(k)}}$ will be larger than $D_0$. As a result, with a learning rate $\mu$, the subsequent $\theta^{(k)}$ updated as $\theta^{(k)} = \theta^{(k)} - \mu \frac{\partial \mathcal{L}}{\partial \theta^{(k)}}$, will also have a rank greater than $D_0$​. This effectively resolves the rank constraint issue. Furthermore, this is also supported by the empirical evidence supporting as presented in Figure 4, where IDInit's success in surpassing the rank of $D_0$​ is demonstrated.
>
> We believe that this additional explanation clarifies the theoretical foundation of our approach and addresses the concerns raised. We will revise the proof in the revised manuscript to ensure a comprehensive understanding of our methods and findings.
>
>
> > The authors claim that the rank constraint can be broken by IDI even when non-linearity like ReLU is not applied. It seems contradict approximation theory which emphasizes the importance of non-linearity to ensure expressivity. It would be great for authors to provide more insights on this point.
>
> We appreciate the reviewer's point regarding the possible contradiction between IDInit and the approximation theory. To clarify, our claim is that IDInit has the capacity to break the rank constraint even without traditional non-linearities such as ReLU. This does not negate the significance of non-linearities in enhancing expressivity, as emphasized by approximation theory. In more detail, our work demonstrates that IDInit's mechanism allows for an increase in the rank of learned representations through padding the identity matrix repeatedly., independent of non-linear transformations. When non-linearities like ReLU are introduced, as per approximation theory, they indeed augment the expressivity of the IDInit. In summary, IDInit, in conjunction with non-linear activations, provides a compounded benefit to the model's expressivity. Therefore, IDInit's impact is additive to the benefits conferred by non-linearities, rather than contradictory.
>
> > Authors mention that dead neurons problem happens when batch normalization set to 0 or downsampling operation cause 0 filled features. However, these cases are not common in practice and it's better to motivate more on why IDIZ is important.
>
> Thank you for the comment on IDIZ in addressing the issue of dead neurons, particularly in relation to batch normalization (BN) and downsampling operations. However, we contend that the scenarios we described are common in practice.
>
> - Batch Normalization: The practice of setting the gamma parameter of the last BN layer in a ResNet block to zero is not only recommended for enhanced performance as indicated in references \[1\]\[2\]\[3\], but also a default setting in widely-used packages like timm \[4\]. This approach, while improving performance, can inadvertently lead to dead neurons, a problem IDIZ aims to mitigate.
>
> - Downsampling: In common downsampling practices, there are two prevalent approaches: (i): Using pooling (e.g., avgpool) to reduce feature resolution, coupled with zero-padding to expand channel size (as shown in Line 40 of \[5\] and option 'A' in Line 66 of \[6\]). (ii): Employing convolution to simultaneously reduce resolution and expand channel size (evident in Line 241 of \[7\] and option 'B' in Line 72 of \[6\]). In particular, for the widely-used datasets like CIFAR-10, option (i) is frequently preferred.
>
> IDIZ is designed to generalize the utilization of IDInit by solving dead neurons of above conditions which is also common settings. So that IDIZ can be used more widely for addressing the disharmonious nature of identity-control methods, which avoids potential risks in compatibility with other techniques. We believe this is important characteristic to an initialization algorithm.
>
> We will revise our manuscript to better discuss these points, ensuring that the relevance and utility of IDIZ are clearly and concisely presented.

---

> > ### Author Response · Authors · 2023-11-20
> > **Response to Reviewer FbDy (2/3)**
> >
> > > What is the meaning of zero down-sampling in Table 2, is this a special downsampling operation compared to standard downsampling (like avgpooling) in ResNet?
> >
> > Sorry for the unclear description of the term "zero down-sampling" in Table 2 of our manuscript. To clarify, this term refers specifically to the first downsampling setting (i) mentioned in our last response. This approach is commonly used for datasets like CIFAR-10 and involves the use of pooling (e.g., avgpool) to reduce the resolution of features, coupled with zero-padding to expand the channel size. In this case, ZerO performs poorly since affected by dead neurons.
> >
> > To enhance the comparative analysis, we utilize ResNet-18 as the backbone, ensuring the avoidance of dead neurons for ZerO. Specifically, we implement 1×1 convolution for downsampling in ResNet-18, as detailed in \[6\], adhering to the downsampling setting (ii). This approach effectively avoids the dead neuron issue, thereby facilitating normal functioning of ZerO. Results are shown in STable 1 below.
> >
> > |Model|Accuracy|
> > |---|---|
> > |ZerO|94.81 $\pm$ 0.05|
> > |IDInit|95.08 $\pm$ 0.13|
> >
> > The results suggest that IDInit still outperforms ZerO of which the dead neurons are solved, which showcases the consistent robustness and effectiveness of IDInit across different downsampling strategies.  We will revise the description for better understanding.
> >
> > > Insufficient explanation on why momentum is important to solve convergence problem of IDInit. It would be great to provide some theoretical insights to support this factor.
> >
> > This is a great question. We would like to give the interpretation from the angle of theory. According to a study on the convergence problem with identity initialization [8], when layers are initialized with the identity matrix, all the layers will be the same at each step until the end, which causes layers not to converge to a ground truth whose eigenvalues contain negative values.  To address this, we delve into the mechanics of the SGD optimizer in our detailed proof presented in Appendix F.2 of the uploaded revision PDF.
> >
> > In the proof, we examine how SGD facilitates differentiation between layers. Specifically, SGD updates weights through a component expressed as:
> >
> >  $$
> > -\eta X_2^TX_2X_1^TX_1 + \eta X_2^TX_2X_1^TY_1 - X_2^TY_2,
> >  $$
> >
> > where $\eta$ represents the learning rate, and $\{X_1, Y_1\}$ and $\{X_2, Y_2\}$  are training pairs sampled from the same dataset $\mathcal{D}$. This update mechanism inherently encourages layers to become asymmetric and distinct from each other, thereby mitigating the issue of identical layers post-training.
> >
> > By amplifying the divergence between layers, momentum plays a critical role in this context . It enhances the effect of SGD's layer differentiation mechanism, thereby significantly addressing the convergence problem. Momentum essentially accelerates the process of breaking the symmetry in layer weights, which is pivotal for effective convergence in networks initialized by IDInit.
> >
> > We believe this theoretical insight, backed by the proof in Appendix F.2, clearly articulates the importance of SGD with momentum in solving the convergence challenge associated with IDInit. We will revise the manuscript to explicate this point more thoroughly to provide a better understanding of the interplay between momentum, SGD, and the layer weights in the context of IDInit.

---

> > > ### Author Response · Authors · 2023-11-20
> > > **Response to Reviewer FbDy (3/3)**
> > >
> > > > It would be great to compare IDI, IDI with loose condition, and IDIZ together to show the effectiveness of IDIZ.
> > >
> > > We appreciate the reviewer’s suggestion to provide a comparative analysis of IDI, IDI with a loose condition, and IDIZ. In Section 4.4 of our manuscript, we initially discussed some of these components individually. Based on the reviewer’s feedback, we have expanded this analysis to further exhibit the interaction of these components when combined.
> > >
> > > STable 2. Analysis of components.
> > >
> > > |          |       Setting 1       |       Setting 2       |       Setting 3       |       Setting 4       |       Setting 5       |      Setting 6       |       Setting 7       |       Setting 8       |
> > > |:--------:|:---------------------:|:---------------------:|:---------------------:|:---------------------:|:---------------------:|:--------------------:|:---------------------:|:---------------------:|
> > > |  Loose   |                       |                       |                       |       &#10004;        |                       |       &#10004;       |       &#10004;        |       &#10004;        |
> > > |   IDIC   |                       |                       |       &#10004;        |                       |       &#10004;        |                      |       &#10004;        |       &#10004;        |
> > > |   IDIZ   |                       |       &#10004;        |                       |                       |       &#10004;        |       &#10004;       |                       |       &#10004;        |
> > > | Accuracy | 86.12$_{±0.52}$ | 92.68$_{±0.08}$ | 89.47$_{±0.24}$ | 87.01$_{±0.29}$ | 92.95$_{±0.21}$ | 92.9$_{±0.18}$ | 90.43$_{±0.14}$ | 93.22$_{±0.05}$ |
> > >
> > > The extended analysis is presented in the newly added STable 2. This table clearly illustrates that while the components 'Loose', 'IDIC', and 'IDIZ' each contribute to improvements independently, their combined application yields the most substantial results. Notably, IDIZ stands out by significantly enhancing accuracies to over 92%. This improvement is primarily attributed to IDIZ’s ability to resolve the dead neuron issue, which is a critical factor in optimizing neural network performance.
> > >
> > > This collective analysis underscores the individual and combined efficacy of the IDInit techniques. The results firmly establish that each component plays a vital role in the overall performance, with IDIZ being particularly instrumental in achieving higher accuracy rates. Our manuscript will be updated to include this comprehensive comparative analysis, ensuring a clear demonstration of the effectiveness and interplay of IDI, IDI with a loose condition, and IDIZ.
> > >
> > > [1] Goyal, Priya, et al. "Accurate, large minibatch sgd: Training imagenet in 1 hour." _arXiv preprint arXiv:1706.02677_ (2017).
> > >
> > > [2] He, Tong, et al. "Bag of tricks for image classification with convolutional neural networks." _Proceedings of the IEEE/CVF conference on computer vision and pattern recognition_. 2019.
> > >
> > > [3] URL: https://github.com/tensorpack/tensorpack/issues/420
> > >
> > > [4] URL: https://github.com/huggingface/pytorch-image-models/blob/main/timm/models/resnet.py
> > >
> > > [5] URL: https://github.com/hongyi-zhang/Fixup/blob/master/cifar/models/resnet_cifar.py
> > >
> > > [6] URL: https://github.com/akamaster/pytorch_resnet_cifar10/blob/master/resnet.py
> > >
> > > [7] URL: https://github.com/pytorch/vision/blob/main/torchvision/models/resnet.py
> > >
> > > [8] Bartlett, Peter, et al. "Gradient descent with identity initialization efficiently learns positive definite linear transformations by deep residual networks." _International conference on machine learning_. PMLR, 2018.

---

> > ### Comment · Reviewer_FbDy · 2023-11-22
> >
> > Thanks for author's detailed response. However, I still have questions about your response on Theorem 3.1.
> >
> > => "Firstly, it is important to understand that the rank of $\theta^{(0)}$ exceeding $D_0$ directly leads the distinct values of the feature $x^{(k)}{ }_{k=1}^{L-1}$ to be larger than $D_0$. Then, since the gradient of $\theta^{(k)}$ is calculated as $\frac{\partial \mathcal{L}}{\partial \theta^{(k)}}=\frac{\partial \mathcal{L}}{\partial x^{(k)}} \circ x^{(k-1)}$ for $k \in 1,2, \ldots, L-2$, the rank of $\frac{\partial \mathcal{L}}{\partial \theta^{(k)}}$ will be larger than $D_0$."
> >
> > I think this is still incorrect. First, the rank of $\theta^{(0)}$ can not exceed $D_0$ as it has the shape of $D_h \times D_0$ (where $D_h > D_0$). Second, I want to kindly remind the authors that number of distinct values of the feature is not strictly correlated with the rank of the weight derivative matrix.
> >
> > For example, considering we have $N$ input vectors $x^{0} \in D_0$, we know the rank of $span(x^{0,1}, ..., x^{0,N})$ is smaller or equal to $D_0$. For $x^{1} = \theta^{(0)} x^{0}$, even when $\theta^{(0)}$ has full rank $D_0$, the rank of $span(x^{1,1}, ..., x^{1,N})$ is still smaller or equal to $D_0$, due to the nature of linear transformation.
> > Now we analysis $\frac{\partial \mathcal{L}}{\partial \theta^{(k)}}= \sum^{N}_{i=1} \frac{\partial \mathcal{L}}{\partial x^{(k,i)}} \circ x^{(k-1,i)}$ (assuming a full-batch gradient descent, you were using $\frac{\partial \mathcal{L}}{\partial x^{(k)}} \circ x^{(k-1)}$ which is a single-batch GD and the matrix is rank-1 only).
> >
> > Take $x^{1}$ as a demonstration and assume $D_L > D_0$ without losing generality. We know the rank of $\frac{\partial \mathcal{L}}{\partial \theta^{(1)}}$ can not exceeed $D_0$ given that
> > the rank of $span(x^{1,1}, ..., x^{1,N})$ is smaller or equal to $D_0$.
> >
> > In other words, $\frac{\partial \mathcal{L}}{\partial \theta^{(1)}}$ consists of $N$ rank-1 matrices, the summation of which can not exceed $D_0$. Thus, I don't agree the authors' claim that "the rank of $\frac{\partial \mathcal{L}}{\partial \theta^{(k)}}$ will be larger than $D_0$".
> >
> > The authors mention the empirical showcase in Figure 4. Is the 3-layer network in Figure 4 a linear network or a non-linear network? It should be a linear 3-layer network if the authors want to demonstrate Theorem 3.1 empirically.

---

> ### Author Response · Authors · 2023-11-22
>
> > In other words, $\frac{\partial \mathcal{L}}{\partial \theta^{(1)}}$ consists of $N$ rank-1 matrices, the summation of which can not exceed $D_0$. Thus, I don't agree the authors' claim that "the rank of $\frac{\partial \mathcal{L}}{\partial \theta^{(k)}}$ will be larger than $D_0$".
> >
> > The authors mention the empirical showcase in Figure 4. Is the 3-layer network in Figure 4 a linear network or a non-linear network? It should be a linear 3-layer network if the authors want to demonstrate Theorem 3.1 empirically.
>
> Thank you for your insightful observations. We acknowledge your point regarding the limitation of $\frac{\partial \mathcal{L}}{\partial \theta^{(k)}}$ to $D_0$. ​However, we maintain that Theorem 3.1 remains valid. In a scenario using full-batch gradient descent for a single round,  $\theta^{(k)}$ indeed retains a rank of at most $D_0$. Nevertheless, when employing Stochastic Gradient Descent (SGD), the weight update process involves iteratively adding gradients of rank $D_0$. As noted in reference [9], the addition of lower-rank matrices can increase the overall rank. This implies that the rank of $\theta^{(k)}$ can be increased, provided the gradients are sufficiently independent, achievable through the use of ample training samples.
>
> To clarify this concept, we utilized a 3-layer network for analysis. Following the notations from Section A.3 in the appendix and assuming $D_h=2D_L=2D_0$​, we demonstrate the process using SGD. After the initial training step with a sample $x_1^{(0)}$​, the gradient is calculated by
> $$\frac{\partial \mathcal{L}}{\partial \theta_1^{(1)}}=\begin{pmatrix*}
> 	\mu\Pi & \mu\Pi  \\\\
> 	\mathbf{0} & \mathbf{0}
> 	\end{pmatrix*},$$
> where $\Pi = \frac{\partial L}{\partial x_1^{(3)}} \circ x_1^{(0)} \in \mathbb{R}^{D_L\times D_0}$. Continuing with a second sample $x_2^{(0)}$​, there is
> $$\frac{\partial \mathcal{L}}{\partial \theta_2^{(1)}}=\begin{pmatrix*}
> (I-\mu\Pi)M(I-\mu\Pi)x_2^{(0)} & (I-\mu\Pi)Mx_2^{(0)}  \\\\
> -\mu\Pi M(I-\mu\Pi)x_2^{(0)} & -\mu\Pi Mx_2^{(0)}
> \end{pmatrix*},$$
> where $M=\frac{\partial \mathcal{L}}{\partial x_2^{(3)}}$. Thus, the residual step can be calculated as
> $$\hat{\theta}^{(1)}=\theta^{(1)}-I=\begin{pmatrix*}
> -\mu\Pi - (I-\mu\Pi)M(I-\mu\Pi)x_2^{(0)} & -\mu\Pi - (I-\mu\Pi)Mx_2^{(0)}  \\\\
> \mu\Pi M(I-\mu\Pi)x_2^{(0)} & \mu\Pi Mx_2^{(0)}
> \end{pmatrix*}.$$
> Without loss of generality, assuming $rank(\Pi)=D_0$, the rank of $\hat{\theta}^{(1)}$ should exceed $D_0$​.
>
> Regarding your query about Figure 4, the network used in this experiment is indeed a linear 3-layer network, as per the requirements to empirically demonstrate Theorem 3.1. There are no non-linear activation functions involved, ensuring the network’s alignment with the theorem's conditions.
>
> We appreciate your feedback and will ensure to include a comprehensive revision and discussion of this aspect in our next manuscript update. We hope this response addresses your concerns effectively.

---

> > ### Comment · Reviewer_FbDy · 2023-11-22
> >
> > Thanks for your clarification. However, there is a fundamental error in your proof: the vector outer product always has rank-1 if two vectors are non-zero. $rank(\prod)$ can not be $D_0$ (if $D_0 > 1$) but 1 at most.

---

> ### Author Response · Authors · 2023-11-22
> **Thank you for the reply**
>
> Thank you for the great insight. Indeed, it is accurate that the rank of the outer product of two non-zero vectors is at most 1. However, for batch data with size $m \ge D_0$, the matrix $\Pi$ can still achieve a rank of $D_0$. This is due to the aggregation of multiple rank-1 matrices arising from each instance in the batch. As these rank-1 matrices are derived from different instances, their cumulative effect can lead to an overall increase in the rank of the aggregated matrix $\Pi$.
>
> Furthermore, we also empirically validate that even with a batch size of 1, through iterative training steps, the rank of $\hat{\theta}^{(1)}$ can exceed $D_0$. This phenomenon occurs due to the successive accumulation of gradients over multiple steps.
>
> Thanks again for your thoughtful question! We are happy to discuss more if you have any other questions.

---

> ### Author Response · Authors · 2023-11-23
> **Updated Proof of Theorem 3.1**
>
> Dear Reviewer FbDy,
>
> We appreciate your feedback and have taken the opportunity to clarify and revise the proof of Theorem 3.1. This revised proof, along with a detailed explanation, is also included in Section A.3 in the appendix in the uploaded revision PDF. In this revised analysis, we focus on batch data rather than single data points.
>
> Assume weights are updated with the stochastic gradient descent (SGD). Without loss of generality, we set $D_h = 2D_0 = 2D_L$. Given two batches of inputs as $\{x_1^{(0, 1)}, x_1^{(0, 2)}, \dots, x_1^{(0, N)}\}\in \mathbb{R}^{D_0}$ and $\{x_2^{(0, 1)}, x_2^{(0, 2)}, \dots, x_2^{(0, N)}\}\in \mathbb{R}^{D_0}$, where $N \ge D_0$ is the batch size. Therefore, the initial gradient of $\theta^{(1)}$ are
> $$
> \frac{\partial \mathcal{L}}{\partial \theta^{(1)}} =
> \begin{pmatrix*}
>     \Pi & \Pi \\\\
>     \mathbf{0} & \mathbf{0} \\
> \end{pmatrix*},
> $$
>
> where $\frac{\partial \mathcal{L}}{\partial \theta^{(0)}} \in \mathbb{R}^{D_h \times D_0}$, $\frac{\partial \mathcal{L}}{\partial \theta^{(1)}} \in \mathbb{R}^{D_h \times D_h}$, $\frac{\partial \mathcal{L}}{\partial \theta^{(2)}} \in \mathbb{R}^{D_L \times D_h}$, $\Pi = \frac{1}{N}\sum^N_{i=1}\frac{\partial L}{\partial x^{(L)}} \circ x_1^{(0, i)} \in \mathbb{R}^{D_L\times D_0}$, and $\mathbf{0}$ is a zero values. $\circ$ denotes outer production.
>
> After training with the second data batch, the gradient is calculated as follows:
> $$
> \frac{\partial \mathcal{L}}{\partial \theta^{(1)}}=
> \begin{pmatrix*}
> (I-\mu\Pi)M(I-\mu\Pi)K & (I-\mu\Pi)MK  \\\\
> -\mu\Pi M(I-\mu\Pi)K & -\mu\Pi MK
> \end{pmatrix*},
> $$
> where $M=\frac{\partial \mathcal{L}}{\partial x_2^{(3)}}$ and $K=\frac{1}{N}\sum^N_{i=1}x_2^{(0, i)}$. This leads to the following residual component:
> $$
> \hat{\theta}^{(1)} =
> \theta^{(1)} - I =
> \begin{pmatrix*}
> -\mu\Pi - \mu(I-\mu\Pi)M(I-\mu\Pi)K & -\mu\Pi - \mu(I-\mu\Pi)MK  \\\\
> \mu^2\Pi M(I-\mu\Pi)K & \mu^2\Pi MK
> \end{pmatrix*},
> $$
> Without loss of generality, assuming $rank(\Pi)=D_0$, we can conclude
> $$
>     rank(\hat{\theta}^{(1)}) \ge D_0.
> $$
> Therefore, IDInit can break the rank constraint by achieving the rank of $\hat{\theta}^{(1)}$ larger than $D_0$.
>
> We sincerely hope that this revision addresses your concern. If there are any further questions or clarifications needed, please do not hesitate to let us know.
>
> Kind regards,
>
> Authors

---

### Author Response · Authors · 2023-11-20
**General Response**

We deeply appreciate the insightful and considerate feedback provided by all of the reviewers. We are glad that all the reviewers (FbDy, eQMM, and GXx1) consider the identity initialization to be an interesting and novel method, and think the proposed IDInit to be validated through sufficient experiments. Furthermore, we appreciate Reviewer FbDy's commendation of our manuscript's clarity and coherence, Reviewer eQMM's positive evaluation of our technical contributions and theoretical analysis, and Reviewer GXx1's affirmation of IDInit's practical benefits, including its deterministic nature and ease of implementation. Additionally, we have undertaken our best efforts to comprehensively address all the queries raised by the reviewers, which included conducting further experimental comparisons and providing detailed clarifications. The extra theoretical and empirical results have been provided and are readily accessible within the uploaded revision PDF document.

---

### Meta-Review · Area_Chair_jDGQ · 2023-12-21

**Metareview:**

This paper proposes a novel initialization method for neural networks designed to improve performance and convergence during training. The reviewers are divided, with the potential utility and simplicity of the proposed method being praised, but with significant concerns about Theorem 3.1 and the patch-maintain scheme. After discussion with the reviewers (including private iteration with FbDy and GXx1) and discussion with the Senior Area Chair, it seems like concerns persist (see message below from FbDy). I therefore must recommend rejection and suggest the authors consider these points and revise (either in content or exposition) in a future version.

- - -

Identical input features (in their case, copying the input features to span over entire hidden dimension) do not promote feature diversity, which prevents rank from growing. I can not verify the correctness of the last comment by author as it still clarification on the last equation to conclude that the rank is strictly larger than D_0. It seems the authors not fully understand Theorem 3.1 given the errors I pointed out about computing gradients given batched gradient descent and the rank of vector outer product. Thus I would like to remain my score and recommendation.

**Justification For Why Not Higher Score:**

See meta-review.

**Justification For Why Not Lower Score:**

n/a

---

### Decision · Program_Chairs · 2024-01-16

Reject